Pathogen avoidance and prey discrimination in
ants. *R. Soc. open sci.* **7**: 191705.

Subject Areas:
behaviour/ecology

Keywords:
avoidance, scavenging, entomopathogenic fungi,
ants, sanitary strategies

Author for correspondence:
Hugo Pereira
e-mail: hpereira@ulb.ac.be

Electronic supplementary material is available
online at https://doi.org/10.6084/m9.figshare.c.
4853166.

# Pathogen avoidance and prey discrimination in ants

## Hugo Pereira and Claire Detrain

Unit of Social Ecology, Université Libre de Bruxelles, CP 231, Boulevard du Triomphe,
1050 Brussels, Belgium

HP, 0000-0001-7421-9396; CD, 0000-0002-3068-1877

Insect societies have developed sanitary strategies, one of which
is the avoidance of infectious food resources as a primary line
of defence. Using binary choices, we investigated whether
*Myrmica rubra* ants can identify prey that has been artificially
infected with the entomopathogenic fungus, *Metarhizium
brunneum*. We compared the ants' foraging behaviour towards
infected prey at three different stages of fungus development:
(i) prey covered with fungal conidia, (ii) prey freshly killed by
the fungus and (iii) sporulating prey. Most foragers retrieved a
corpse covered with a high number of spores but they
consistently avoided a sporulating prey and collected less prey
that had recently died from fungal infection. Furthermore,
ant responses were highly variable, with some individuals
retrieving the first prey they encountered while others
inspected both available prey before making a decision.
Workers were not repelled by the simple presence of fungal
conidia but nevertheless, they avoided retrieving cadavers at
later stages of fungal infection. We discuss how these different
avoidance responses could be related to: differences in the
ants' perceptive abilities; physico-chemical cues characterizing
fungus-infected prey or in the existence of physiological or
behavioural defences that limit sanitary risks associated with
potentially contaminated resources.

## 1. Introduction

Animals living in groups are exposed to higher risks of disease and
transmission of parasites between group members [1]. This is the
case for insect societies where the spreading of pathogens is
facilitated by frequent contacts or food exchanges between closely
related workers that live together in confined nesting sites [1–4].
To counteract parasite pressure, insect societies have developed
effective defence mechanisms at the group level which reduce
sanitary risks [5] and which complement the physiological
defences existing at the level of the individuals. These collective
behavioural defences contribute to the so-called social immunity
of the whole colony: they enable the insect society both to reduce

its level of exposure to sources of infection as well as to increase its resistance to pathogens and parasites (for a review see [6,7]).

During the last decades, much progress has been made in understanding how insect societies develop collective strategies of resistance to parasites [4,6]. In particular, ants exhibit a remarkable variety of behavioural adaptations which ultimately reduce the probability of them getting infected and which lower the pathogen load on nestmates, on stored food or even on nest material. Strategies of resistance to pathogens include the targeted grooming of infected nestmates [8–10], the application of antimicrobial substances on nest material [11,12], the effective undertaking of dead nestmates [13] or the management of waste items [13,14]. Furthermore, socio-spatial strategies of resistance to pathogens, i.e. organizational immunity, [15] aim to segregate and reduce interactions between potentially infectious individuals (e.g. nest-cleaning workers or moribund ants) and vulnerable individuals (e.g. the queen or the brood) [16,17]. While we better understand how insect societies limit disease outbreaks once pathogens have entered the nest or have contaminated nestmates, less is known about their first line of defence against pathogens, that is how they avoid potential sources of contamination from the outset. The detection of pathogens is a fundamental prerequisite for developing efficient strategies of resistance to parasites, with natural selection privileging vigilant hosts. In the case of insect societies, the early and accurate detection of pathogens may be vital since it can mitigate the risk of the disease spreading inside densely populated nests [18]. In particular, in ants species nesting in the ground, workers are regularly exposed to a wide range of microorganisms including some fungal strains that can be life-threatening for the insects [19]. Pathogen avoidance should have evolved due to the expected benefits for colony survival. Nevertheless, the responses of ant colonies to infectious spores are highly variable between colonies of the same species as well as across different ant species. Indeed, colonies of several ant species actively avoid fungal-contaminated surfaces [20,21], whereas some ant colonies or even young founding queens can show a clear preference for fungus-infected nesting sites [22–24]. Furthermore, the level of avoidance can differ quantitatively depending on the concentration and virulence of pathogens [25] but also on some life-history traits of ant species [20].

Likewise, inside the colony itself, one may assume that levels of detection and avoidance of pathogens should differ among colony members, depending on how frequently they encounter potential sources of infection. In this respect, the detection of parasites is expected to be particularly efficient among the worker individuals that explore the nest surroundings, those that inspect unknown food sources and those that retrieve prey items inside the nest. Scouts and foragers are potentially the colony members which are the first and the most frequently exposed to contaminated food or areas. This is particularly true for scavenging ant species [26], since dying prey or insect cadavers commonly host a wide range of entomopathogenic microorganisms. Besides being harmful to the forager itself, the retrieval of an infected prey item inside the nest may expose the whole colony to a high risk of microbial growth and disease. Indeed, food items, such as fungus-infected prey, can sporulate and become a dangerous source of horizontal transmission of parasites inside the nest. Foragers' early assessment of the sanitary risks associated with food items, could act as a first 'filter' which reduces the need for further investment in resistance and/or tolerance strategies at the colony level.

A few studies on eusocial insects have shown that foragers can actually detect and avoid contaminated food. For example, bumblebee foragers are repelled by flowers contaminated with a protozoan flagellate parasite *Crithidia bombi* [27]. In *Atta* ant species, major foragers avoid retrieving leaves carrying a high microbial load or containing fungal endophytes, while small workers clean the retrieved leaf fragments of potential pathogens such as *Metarhizium anisopliae* spores [28–30]. These behaviours contribute to protecting the mutualistic fungal crop from competitive and pathogenic microbes. Likewise, a comparative study on several ant species showed that foragers avoid feeding on food items that were covered with a huge amount of conidia from the entomopathogenic fungus *M. anisopliae* [20]. While these studies demonstrate the ants' ability to detect fungal spores, workers of scavenging ant species are fairly unlikely to encounter dead prey covered with conidia under natural conditions. Indeed, the adhesion of conidia to the insect body represents the first and ephemeral stage of the fungal developmental cycle and lasts around 48 h, until the appressoria allow the penetration of hyphae inside the living host through the cuticle [31].

One may assume that the detection of more 'cryptic' sources of infection is particularly adaptive for animals like ants that do not consume their prey at the discovery site but that retrieve potentially harmful food items inside the nest. Furthermore, these infectious items can be stored for prolonged periods inside their nest and/or shared with individuals that are more vulnerable to diseases, such as larvae. However, we still do not know whether the ability to detect fungal infection in their prey exists for 'cryptic' stages of food contamination, i.e. when all conidia have penetrated the insect body, and mycelium has invaded

the haemocoel. The early detection of fungal contamination is assumed to be highly beneficial to scavenging ants in order to protect them from the risk associated with the retrieval of a fungus-killed insect inside the nest. Likewise, ants would also benefit from being able to avoid 'old' decaying prey out of which the fungus has sporulated, thereby avoiding this immediate threat for the whole colony.

In this study, we investigate whether workers of the red ant *Myrmica rubra* can detect and discriminate between a non-infected prey and a prey infected by the entomopathogenic fungus *Metarhizium brunneum*. We focused on three different developmental stages of the fungal microparasite: the attachment of conidia to the host cuticle, the colonization of the host haemocoel by fungal mycelium and the sporulation of fungus out from the host cadaver. The red ant, *M. rubra*, which has an omnivorous diet with a high scavenging activity [32], was chosen to study pathogen avoidance in a context of prey foraging. By carrying out individual choice tests, we quantified and compared the retrieval behaviour of *M. rubra* foragers which encountered (i) a prey covered with conidia, (ii) a prey recently killed by the fungus when the mycelium had invaded the insect haemocoel but was not yet sporulated, and (iii) a prey killed by the fungus that had sporulated outside the insect body. This study aims to better understand how strategies of pathogen avoidance contribute to social immunity in ants.

# 2. Material and methods

## 2.1. Studied ant species and rearing of ant colonies

*Myrmica rubra* is a polydomous, polygynous and monomorphic ant species that is common in European temperate areas. This ant species lives in semi-humid conditions and can be found in biotopes such as semi-open grasslands or paths' borders. *M. rubra* nests are dug in various substrates such as in ground under stones, inside rotting wood, among roots of nettles and bramble bushes [33]. Queenright colonies of *M. rubra* containing brood, were collected in Belgium from the localities of Sambreville (50° 25′ 59.62″ N;  4° 37′ 22.12″ E),  Falisolle  (50° 25′ 11.99″  N; 4° 37′ 50.41″ E) and Brussels (ULB university campus; 50° 49′ 05.1″ N 4° 24′ 01.6″ E) during the summers of 2017 and 2018. In the laboratory, ants were kept in a room at 21 ± 2°C, around 50% relative humidity and a light/dark period of 12–12 h per day. Each colony was placed in a box (22 × 14 × 9 cm) with walls coated with polytetrafluoroethylene (Fluon; Whitford, UK) to prevent ants from escaping. The nest was composed of a test tube with a water reservoir plugged with a cotton wool and covered with a red filter. We provided the ant colonies with water and sucrose solution (0.3 M) ad libitum as well as with proteins and lipids by giving them mealworms (*Tenebrio molitor*) twice per week.

### 2.1.1. Rearing of fruit flies

Prey used in the choice tests were *Drosophila melanogaster* fruit flies that belonged to the 'vestigial wings' phenotype. These fruit flies were reared on a home-made food substrate (79% apple sauce, 3% oat bran, 12% mashed potatoes in snowflakes, 5% white vinegar) at a room temperature of 21°C and a 50% relative humidity. We dropped raffia strings on the mixture in order to facilitate the pupation of larvae. We renewed the food substrate once per month.

### 2.1.2. Entomopathogenic fungus and prey contamination

We used a commercial strain of *M. brunneum* F52 (formerly known as *Metarhizium anisopliae* var. *anisopliae*) from Novozymes Bayer™. *M. brunneum* is a common generalist entomopathogen that can infect more than 200 insect species and that occurs in the natural habitat of *M. rubra* nests [34]. This entomopathogen is commonly used to investigate how insect societies can develop sanitary strategies against pathogens. Once in contact with the host body, *M. brunneum* conidia attach themselves on its surface and develop appressoria that pierce the cuticle with fungal enzymes. The fungal mycelium then spreads inside the host haemocoel and eventually causes the death of the insect. Several days after the insect's death, the fungus starts sporulating outside the corpse and releasing new conidia in the environment.

Depending on the developmental stage of the fungus that we wanted to study, we used two different methods to contaminate flies. To obtain a prey freshly dead from fungal infection or a sporulating prey, we exposed living fruit flies to *M. brunneum* conidia by vortexing them together with a sporulating corpse, four times during 5 s at a speed of 1500 r.p.m., as described in the protocol of Leclerc and Detrain [35]. Then, we waited for the death of contaminated flies, which occurred around 5 days after their exposure to conidia. The day of their death, prey were either immediately used for testing ant

responses (i.e. freshly fungus-killed prey) or were allowed to sporulate thanks to the following procedure. To obtain sporulating flies, we washed their corpses following the standard method of Lacey [36] to prevent the growth of opportunistic microorganisms on the cadaver. Then, we placed fly corpses on a wet filter paper inside a closed Petri dish, keeping them in a thermostatic cabinet at 25°C. This enabled the emergence of conidiophores on the prey cuticle during the next 2 days.

Another procedure was followed to obtain a prey whose body was covered with a known amount of spores. We topically applied 1 µl of a Tween 20 suspension of conidia on the cuticle of a fly that was freshly killed at −10°C and we let the solvent evaporate out of the prey body surface for 1 h. The suspension of conidia was made by putting a sporulating fly in 500 µl of a Triton X-20 solution (0.05%) that was centrifuged in an Eppendorf for 5 min at 6000 r.p.m. We cautiously removed the supernatant and added 500 µl of Tween 20 (0.05%). We scattered the pellet and homogenized the solution by shaking the Eppendorf during three phases of 10 s with a vortex mixer. We used a microscope with a Thoma's cell to estimate the concentration of this initial suspension of conidia and we diluted it with Tween 20 in order to reach the desired concentration of spores (low or high amount of spores, see below).

Preliminary experiments showed that the application of Tween 20 on the prey fly had no impact on their retrieval by ants. Indeed, when being offered two freshly dead flies that were impregnated ($n = 152$) or not ($n = 128$) with 1 µl of Tween 20, ants retrieved 69.1% and 72.7% of flies, respectively ($\chi^2$-test: $\chi^2 = 0.27$, d.f. = 1, $p = 0.6$). Likewise, we found no difference in the number of antennal contacts with prey (Wilcoxon test: $V = 20624$, $p = 0.27$) nor in the duration of these contacts (Wilcoxon test: $V = 23950$, $p = 0.1$) between these two conditions. This confirmed that the application of Tween 20 on the prey body did not affect their retrieval by ants, which remained as high as for control flies.

## 2.2. Experimental procedure

We tested in total, 10 queenright colonies that hosted 200 workers and around 50 larvae. In order to motivate foragers to retrieve a prey, we starved colonies before the experiment by depriving them of food for 3 days.

The experimental set-up was made of a circular arena (7.5 cm in diameter, 0.8 cm height) with one entry (2.5 cm width) that opened into the foraging area of the tested colony and that was located at 15 cm from the nest entrance. Binary choice experiments consisted of placing two dead flies at the centre of the experimental arena, each prey being separated by 1 cm. Once an ant entered the arena, the access was closed with a circular barrier in order to prevent other workers from interfering with the tested individual. The experiment started as soon as the ant contacted with their antennae one of the two prey. To be sure that no worker was tested several times, we removed the individual from the colony at the end of the test and we put it in a separate plastic box. After each experimental session, we readjusted the colony size to 200 workers by introducing new ants from the mother colonies.

We tested six choice conditions in which one fly was a control and the other fly differed by its level of infectiousness as well as by the time elapsed since its death.

The behaviour of ants was quantified for the following conditions:

(1) Control prey (Ctrl) versus control prey (Ctrl): Two freshly dead flies that were cold-killed by keeping them at −10°C for 3 min. All control prey were used for tests at most 6 h after their death. This experiment allowed to check for possible spatial bias in prey choice. We tested 128 ants in this condition.

(2) Control prey (Ctrl) versus prey covered with a low amount of conidia (LC): The prey was cold-killed 1 h before the experiment and were impregnated with 1 µl of Tween 20 for the control prey or with a Tween 20 suspension containing a low concentration of conidia ($5 \times 10^6$ spores ml$^{-1}$) for the infected prey. The body of infected prey was thus covered by around 5000 conidia. A total of 110 ants were tested in this condition.

(3) Control prey (Ctrl) versus prey covered with a high amount of conidia (HC): We followed the same procedure as described above but the infected prey was impregnated with 1 µl of a Tween suspension containing a high concentration of conidia ($10^9$ spores ml$^{-1}$) which corresponded to $10^6$ spores deposited on the fly. A total of 60 ants were tested in this condition.

(4) Control prey (Ctrl) versus fungus-killed prey (FKill): The infected prey was a fly that had just died from fungal infection after having undergone the vortexing protocol and that was immediately used for testing on the day of its death. This allowed to see whether ants could early detect fungal infection, even when the prey corpse was not sporulating yet. At the end of the experiment, we

checked that the tested prey actually died from fungal infection by placing it at 25°C under high humidity conditions and by looking for the emergence of conidiophores on the prey cuticle. We discarded from the analysis the experiments for which the FKill fly did not sporulate (13% of tested flies). A total of 132 ants were tested and analysed in this condition.

(5) Control prey (Ctrl) versus sporulating prey (Spo): The sporulating prey was a fly that underwent the vortexing protocol and, once dead, that was sporulating in a thermostatic cabinet for around 2 days (as detailed in the previous section). Before using the sporulating fly in binary choices, we checked with a binocular microscope the presence of conidiophores on the cuticle of the insect cadaver [37,38]. A total of 75 ants were tested in this condition.

(6) Decaying prey (Dc) versus sporulating prey (Spo): The sporulating fly was obtained as described above. The decaying prey followed the same treatment as the sporulating one but was killed by exposure to cold and had no contact with fungal conidia. The decaying fly was used as a control specific for the sporulating prey in order to assess to what extent the fungal sporulation itself or the general decaying process influenced prey retrieval by foragers. A total of 136 ants were tested in this condition.

All the experiments (except for the Ctrl–HC condition) were carried out from October 2017 till March 2018 by using foragers originating from six colonies. The last experimental condition (Ctrl–HC) was performed with foragers originating from four new colonies during April 2019. Before analysing together these two experimental series, we checked that ants from the first six colonies did not differ from the other four colonies in their response to the same pathogenic threat (i.e. either the Ctrl–LC ($n = 60$ tested ants) or the Ctrl–Spo condition ($n = 20$ tested ants)). We found that the retrieval rate of infected prey did not differ between the two experimental series for the Ctrl–LC condition: ($\chi^2$-test: $\chi^2 = 0.53$, d.f. = 1, $p = 0.46$) nor for the Ctrl–Spo condition ($\chi^2$-test: $\chi^2 < 0.001$, d.f. = 1, $p = 1$; electronic supplementary material, table S1). Moreover, ants were as eager to touch prey, with a similar percentage of ants contacting one or both prey ($\chi^2$-test: for Ctrl–LC condition: $\chi^2 = 1.10$, $p = 0.29$; for Ctrl–Spo condition: $\chi^2 = 0.19$, $p = 0.66$) and with a similar time spent inspecting the infected prey between the two experimental series (Mann–Whitney: for Ctrl–LC condition: $U = 3746.5$, $p = 0.14$; Ctrl–Spo condition: $U = 830.5$, $p = 0.47$; electronic supplementary material, table S1). Put all together, this indicates that ants did not differ in their level of pathogen avoidance when being tested in the first experimental series of six colonies or in the second series of four colonies. In total, we tested 641 workers in individual trials coming from 10 different colonies (electronic supplementary material, table S2).

### 2.2.1. Behavioural measures

Once the ant entered the test arena and contacted one of the two prey with its antennae, we video recorded its behaviour for 5 min by using a webcam (Logitech Pro HD C920). We considered a prey to be chosen by the tested ant, once the worker transported the fly until the entrance of the arena. If no retrieval occurred during the allocated 5 min, we considered that the individual did not make a choice. We also measured several behaviours displayed by ants around prey such as (i) the number of antennal contacts, (ii) the time spent contacting each prey, (iii) the latency time before making their choice, and (iv) the number and duration of self-grooming.

## 2.3. Statistical analysis

All data were analysed with R software (v. 3.5.0) and all tests were two-tailed with a significance level of $\alpha = 0.05$. No data met normality conditions. We used generalized linear mixed models (GLMMs) from the 'lme4' R-package, to analyse data that met the models' assumption and that showed no overdispersion based on model deviance/degrees of freedom values. The colony from which the tested individuals were obtained, was considered as a random effect and the significance of fixed effects was tested using the likelihood ratio test (LRT) between one model containing the effect and one without it. When data could not be transformed to meet the model's assumptions (e.g. underdispersion), we ran non-parametric tests.

We analysed separately the group of ants that came into contact with just one prey and the group of ants that touched both prey before making their choice.

For the ants that contacted only one prey, we analysed the rate of prey retrieval by using GLMMs with a logit link function and by considering the level of prey fungal infection as a fixed effect. When significant, we performed post hoc $\chi^2$ pairwise comparisons with a Bonferroni correction. We used Kruskal–Wallis tests to check whether the number and the duration of antennal contacts on prey

differed depending on its level of fungal infection. When significant, we performed Dunn's *post hoc* tests with Bonferroni correction using the 'FSA' package. We also used $\chi^2$-tests to assess homogeneity in the percentage of ants that groomed themselves after having contacted prey that differed by its fungal infectiousness. We carried out a Cox proportional hazards regression model by using the R-package 'Survival' to test differences in the latency to retrieve a prey depending on the infectious level of the contacted fly. In order to facilitate data interpretation, we computed hazard ratios (HR) by exponentiating the parameter estimates of the Cox models. These HR accounted for the probability of an event (i.e. the retrieval of a prey) to occur in an experimental condition divided by the probability of the same event to occur in the reference condition (here the Ctrl–Ctrl condition). Individuals that did not make a choice were integrated in the database by being assigned the maximal duration of observation, i.e. 300 s.

For the individuals that came into contact with both prey, we analysed the rate of prey retrieval by using GLMMs with a logit link function and by considering the experimental condition as a fixed effect. When significant, pairwise comparisons of retrieval rate between conditions were performed using $\chi^2$-tests with a Bonferroni correction. Furthermore, for the ants that made a choice, we tested, in each condition, whether they showed a preference for one of the two prey by using an exact binomial test with a probability of 0.5. Indices of contact occurrence and contact duration were calculated to assess a differential avoidance of the two prey by the foragers. For each tested ant, we subtracted the number or duration of contacts made on the control prey from the value observed on the contaminated prey. Hence, a negative value of the index reflected an avoidance of the contaminated prey by ant workers. In each condition, a Wilcoxon signed-rank test was used to compare these indices to the theoretical value zero, which represented an equal interest for both prey. The percentage of self-grooming ants and the duration of groomings were compared across conditions by using a $\chi^2$-test and a Kruskal–Wallis test, respectively. Finally, for the ant individuals that contacted both prey, we checked whether their decision to retrieve the control prey was altered by the presence of the nearby infected fly. To this aim, we performed a Cox proportional hazards regression model to compare the latency time to retrieve the control prey between the different experimental conditions. All data values were expressed as medians and the first and third quartiles (median [Q1,Q3]). All the figures were achieved with the package 'ggplot2'.

# 3. Results

The overall percentage of ants that decided to retrieve a prey differed across the experimental conditions (GLMM: LRT, $\chi^2 = 59.83$; d.f. = 5; $p < 0.001$; figure 1). A majority (73%) of the tested ants took a prey when two freshly dead flies were offered (Ctrl–Ctrl condition). Likewise, 70% of foragers retrieved a prey item when a low amount of conidia covered one of the two flies (Ctrl–LC condition). The overall percentage of ants retrieving a prey decreased to 53% when one of the flies had its body covered with a high amount of conidia (Ctrl–HC condition) and to 60% when one just died from fungal infection (Ctrl–FKill condition). This decreased retrieval of prey became even more pronounced as soon as workers faced a binary choice that included a sporulating prey (i.e. the Ctrl–Spo condition and the Dc–Spo condition). In these two latter cases, the percentage of ants taking a prey dropped to 35% and 39%, respectively, and were significantly lower than those obtained for all the other experimental conditions ($\chi^2$ pairwise comparisons: all $p$-values < 0.05) except when comparing with the Ctrl–HC condition ($\chi^2$ pairwise comparisons: Ctrl–HC versus Ctrl–Spo: $p = 1$; Ctrl–HC versus Dc–Spo: $p = 0.68$; electronic supplementary material, table S3).

We found that several ant individuals limited their foraging effort to contacts with only one of the two prey, i.e. the first fly they encountered after having entered the test arena. Other ants were eager to explore all the available resources and came into contact with both prey. Thereafter, those foragers either chose to take one prey or they returned unloaded to the nest or they were still inspecting the flies at the end of the allocated observation time. This suggested that individual foragers may differ in their level of exploration of food opportunities as well as in their propensity to make a decision about prey retrieval. We investigated whether such an inter-individual variability in the foraging behaviour may depend on the prey. First, we found that the percentages of ants that contacted only the first encountered prey differed between the experimental conditions (GLMM: LRT, $\chi^2 = 42.10$; d.f. = 5; $p < 0.001$; figure 2). For most of the experimental conditions (Ctrl–Ctrl, Ctrl–LC, Ctrl–FKill), around half of the tested individuals contacted a single fly without even inspecting the other prey item. However, when one of the two flies was covered with a high amount of conidia (Ctrl–HC condition), the percentage of ants that contacted only one prey without touching the second fly decreased to 33%. Likewise, the percentage of ants contacting a single

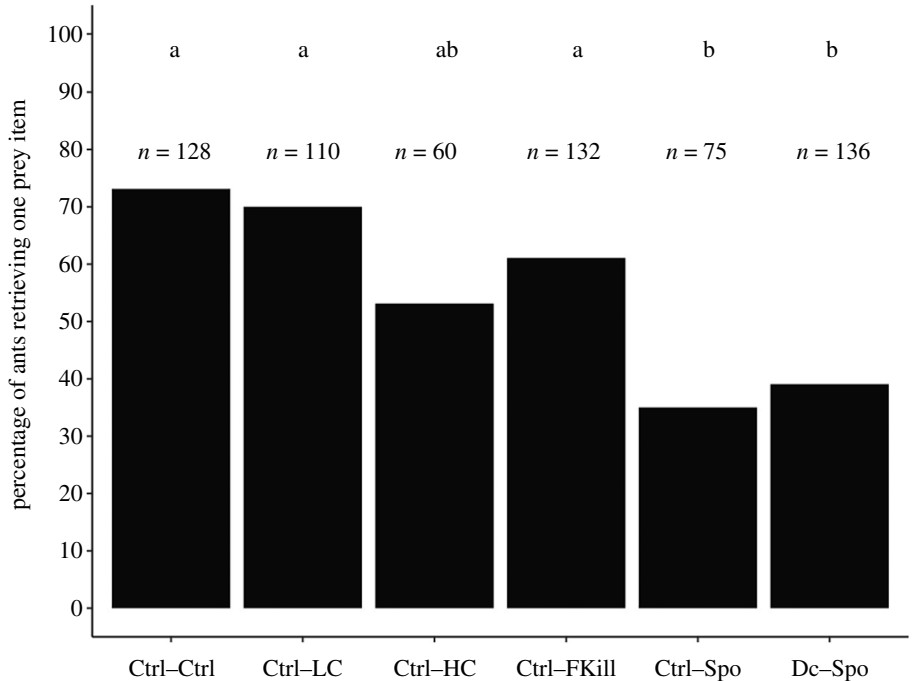

**Figure 1.** Overall percentage of ants retrieving a prey item for the different experimental conditions. Number over bar indicates sample size. *Post hoc* multiple pairwise comparisons were made with Bonferroni correction. Experimental conditions sharing a common letter were not significantly different.

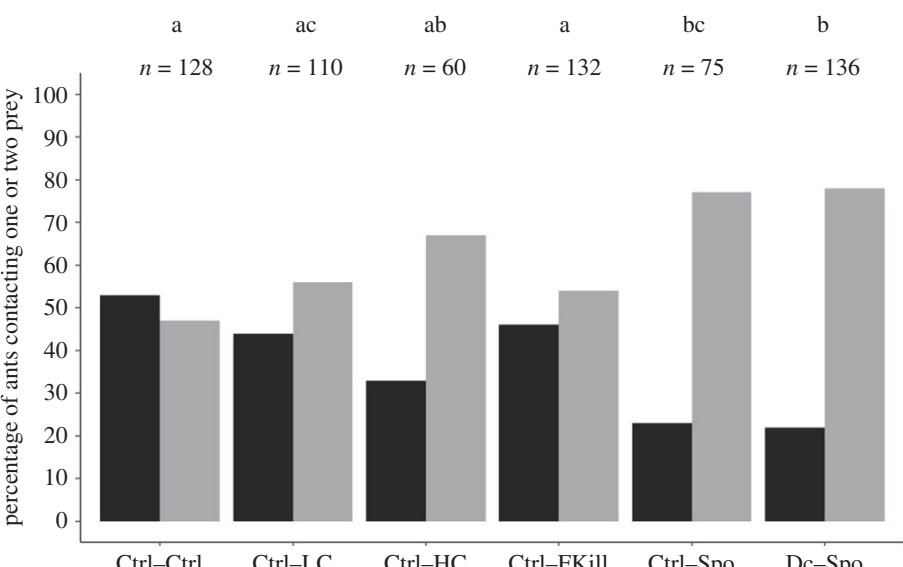

**Figure 2.** Percentage of ants contacting one prey (black bars) or both prey (grey bars) depending on the experimental condition. *n* value above bar is the total number of ants that contacted the corresponding prey. *Post hoc* multiple pairwise comparisons were made with Bonferroni correction. Experimental conditions sharing a common letter were not significantly different.

prey became significantly lower and dropped to 22% in the presence of a sporulating prey ($\chi^2$ pairwise comparisons: all *p*-values < 0.05, except when comparing Ctrl–LC versus Ctrl–Spo condition: $p = 0.082$; Ctrl–HC versus Ctrl–Spo: $p = 1$ and Ctrl–HC versus Dc–Spo condition $p = 1$; electronic supplementary material, table S4). Antennal contacts with a sporulating corpse thus made ant workers more eager to inspect alternative prey sources before making a foraging decision.

The foraging choices made by the ants that came into contact with only one prey and those that encountered both prey were analysed separately. For the first group, we examined whether the ants that limited their exploration to a single source exerted a less efficient sanitary control on the retrieved

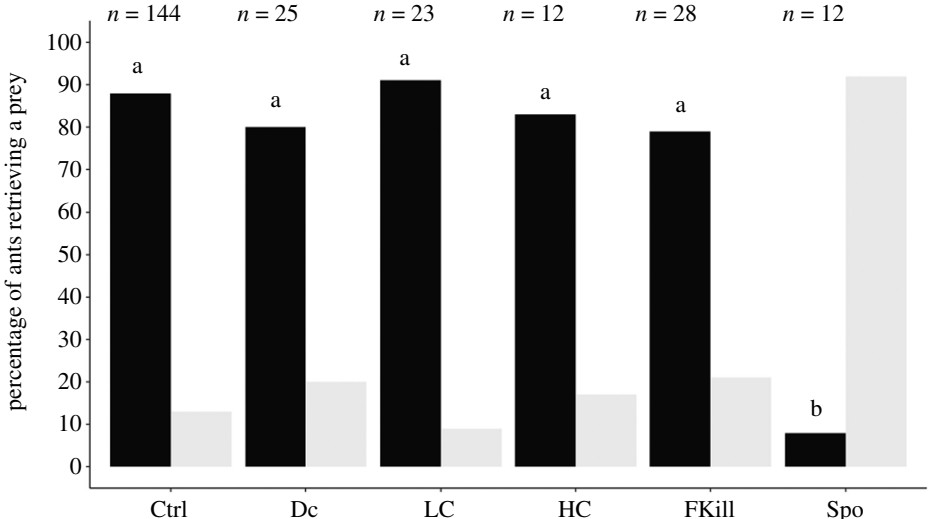

**Figure 3.** Percentage of ants retrieving a prey item (black bars) or not (grey bars) among workers that contacted a single fly depending on its level of fungal infection. For the Ctrl prey, we pooled all the retrieval events observed across the experimental conditions that used a cold-killed control prey ($n = 144$). $n$ value above bar is the total number of ants that contacted the corresponding prey. *Post hoc* multiple pairwise comparisons were made with Bonferroni correction. Bars sharing a common letter were not significantly different.

**Table 1.** For ants that contacted a single prey: total duration of contacts (s) and percentage of self-grooming ants depending on the infectiousness of the contacted prey. Values of median, first and third quartiles are given. We performed a Kruskal–Wallis test on the duration of contacts and a Fisher test on the percentage of self-grooming ants to assess the impact of prey infectiousness.

| | control $n = 144$ | decaying $n = 25$ | low conidia $n = 23$ | high conidia $n = 12$ | fungus-killed $n = 28$ | sporulating $n = 12$ |
|---|---|---|---|---|---|---|
| contact duration (median [Q1,Q3]) | 62 [37,117] | 69 [28,262] | 64 [47,192] | 82 [56,255] | 58 [37,210] | 40 [11,254] |
| | Kruskal–Wallis test: $H = 3.8$, d.f. = 5, $p = 0.58$ | | | | | |
| self-grooming ants (%) | 11.80% | 4% | 8.70% | 0% | 14.30% | 41.70% |
| | Fisher test: $p = 0.053$ | | | | | |

food items. For the second group of workers that came into contact with both prey, we examined whether these ants were able to discriminate the sanitary risks associated with food items and to make adaptive foraging choices that prevented the retrieval of infectious food sources.

## 3.1. Ants that contacted one prey

Workers that touched a single fly significantly differed in their propensity to retrieve the prey depending on its level of fungal infection (GLMM: LRT, $\chi^2 = 36.37$; d.f. = 5; $p < 0.001$; figure 3). Around 80–90% of the ants that first contacted a control cold-killed fly or a decaying prey decided to take it back to the nest, without even inspecting the second fly. Quite surprisingly, nearly all the ants (80–90%) were as likely to retrieve a prey covered with a low or with a high amount of fungal conidia as well as a fly recently dead from fungal infection. By contrast, when the ants came into contact with a sporulating prey, the percentage of retrieving individuals sharply decreased to 8.3% and was significantly lower than those observed for the other prey items ($\chi^2$ pairwise comparisons: all $p$-values $< 0.05$; electronic supplementary material, table S5). This low rate of prey retrieval was not coupled with a reduced duration of ant's contacts with the sporulating prey ($m_{spo} = 40$ s [11,254] versus $m_{ctrl} = 62$ s [39,120]). Indeed, worker individuals spent an equal amount of time inspecting the fly regardless of its infectiousness (Kruskal–Wallis test: $H = 3.8$, d.f. = 5, $p = 0.58$, table 1). Ants usually made a single antennal contact with the prey regardless of its fungal infection except for the sporulating fly that was

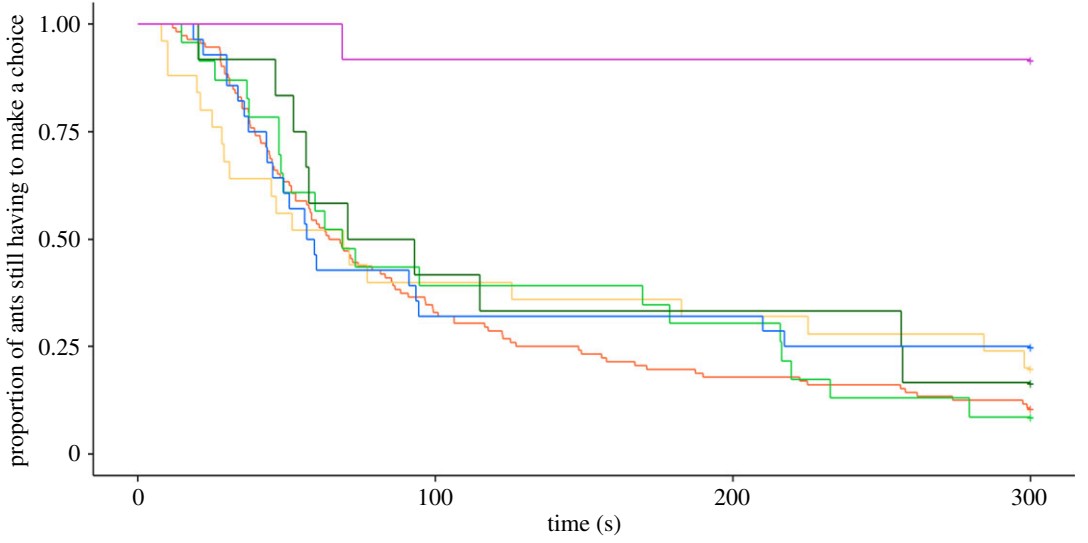

**Figure 4.** Proportion of ants still having to make a choice as a function of the time elapsed since the first contact with a prey. Curves are drawn for the ants that contacted a single prey and differing by its infectiousness. Orange: control prey (Ctrl, $n = 144$), yellow: decaying prey (Dc, $n = 25$), green: prey covered with a low amount of conidia (LC, $n = 23$), dark green: prey covered with a high amount of conidia (HC, $n = 12$), blue: fungus-killed prey (FKill, $n = 28$), violet: sporulating prey (Spo, $n = 12$).

twice contacted by the ants ($m_{Spo} = 2$ contacts [2,3]; Kruskal–Wallis test: $H = 32.9$, d.f. = 5, $p < 0.001$; Dunn's *post hoc* pairwise comparisons: Spo versus all others types of prey all $p$-values < 0.05).

The impact of prey infectiousness on the percentage of self-grooming individuals was nearly significant (Fisher test, $p = 0.053$; table 1). We found that at most 14% of ants groomed themselves after contacting a control prey, a decaying prey, a prey covered with conidia or a prey recently dead from fungal infection (14%). By contrast, 42% of the few workers that had contacted a sporulating fly ($n = 12$) groomed themselves, most probably trying to remove spores from their antennae.

Survival curves revealed that the time elapsed before the ants decided to take a prey differed according to the level of prey infectiousness (Cox model, LRT: $\chi^2 = 35.94$, d.f. = 5, $p < 0.001$; figure 4; table 2). HR of the Cox models allowed us to compare the likelihood of retrieving a prey for an ant tested in a given experimental condition with a reference condition (here the control prey). Overall, individuals were less prone to retrieving a sporulating prey (HR = 0.041, $p = 0.0016$; table 2) than a control fly. On the other hand, the presence of conidia, at low concentration (HR = 0.94, $p = 0.79$) or at high concentration (HR = 0.76, $p = 0.40$) on the fly's body did not hamper its retrieval by ants. Likewise, the time elapsed before the ants decided to retrieve a prey was not altered for prey that had recently been killed by the fungus (HR = 0.78, $p = 0.30$) or a decaying prey (HR = 0.84, $p = 0.47$).

## 3.2. Ants that contacted both prey

The group of ant individuals that came into contact with both prey provided useful information on the ability of foragers to discriminate between prey that differ in their associated risks of fungal infection.

Ants that encountered both prey ($n = 397$) were twice less likely to retrieve food items than the ants having touched a single fly ($n = 244$) (40.3% versus 81.9%; $\chi^2$-test: $\chi^2 = 104.87$, d.f. = 1, $p < 0.001$). For these ants that contacted both prey, the percentage of no-choice differed across experimental conditions (GLMM: LRT, $\chi^2 = 22.96$; d.f. = 5; $p < 0.001$; figure 5). The percentage of no-choice ants was markedly higher when one of the two prey was sporulating than when both prey were freshly dead ($\chi^2$ pairwise comparison: Ctrl–Ctrl versus Ctrl–Spo; $p = 0.039$, electronic supplementary material, table S6) and tended to be higher when one fly was covered with a low amount of spores ($\chi^2$ pairwise comparison: Ctrl–LC versus Ctrl–Spo; $p = 0.068$). In the Ctrl–Ctrl condition, foragers showed no significant preference for retrieving the cold-killed fly that was placed either on the left or on the right side of the test arena (binomial test: $p = 1$, $n = 34$). The foraging choices of ants thus took place irrespective of the location of the flies. In terms of the impact of prey infectiousness, when only considering the ants that decided to take prey, they were as likely to retrieve a fly whose corpse was covered with a low or a high amount of conidia than the control fly (Ctrl versus LC: 59% versus 41%,

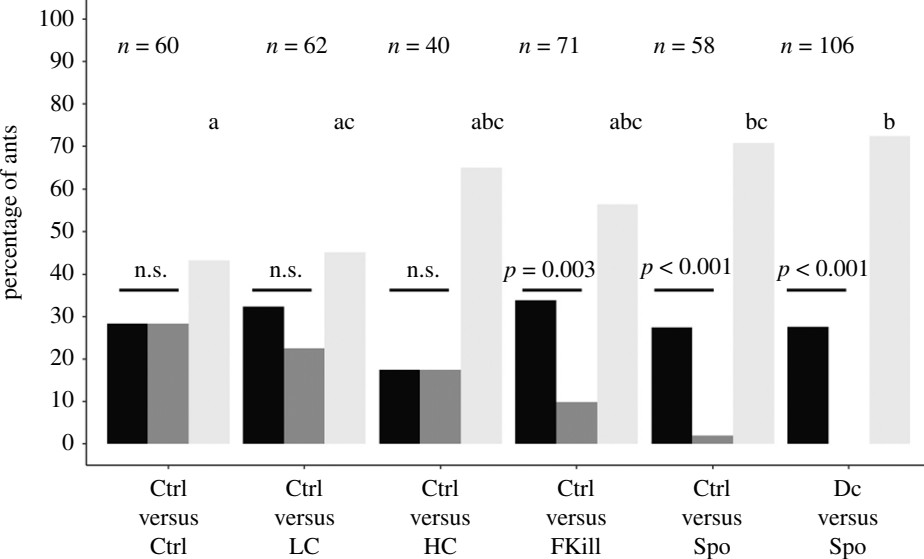

**Figure 5.** Percentage of ants retrieving a prey item depending on the experimental condition, for ants that contacted both prey. Black bars: ants retrieving the control prey (or the decaying prey for the last condition); grey bar: ants retrieving the fungus-infected prey; light grey bars: ants making no choice during the allocated time. Numbers indicate sample size in each experimental conditions. We performed a binomial test to check whether the ants that took a prey showed a preference for one of the two prey. *p*-values of binomial tests are given over bars (n.s.: *p* > 0.05). We also used a GLMM to check the influence of the experimental condition on the percentage of no-choice. *Post hoc* multiple pairwise comparisons were made with Bonferroni correction. Bars sharing a common letter were not significantly different.

**Table 2.** Effect of prey infectiousness on the time elapsed before prey retrieval. Values of estimate parameter, standard error (s.e.), hazard ratio (HR) and confidence interval (CI) of the Cox proportional hazard regression models are given. *p*-Values compare the likelihood of a retrieval event between a given condition and the reference condition (i.e. Ctrl fly).

| variable (reference) | estimate | ±s.e. | HR | 95% CI | *p*-value |
|---|---|---|---|---|---|
| decaying | −0.18 | 0.25 | 0.84 | 0.52–1.36 | 0.47 |
| low conidia | −0.07 | 0.24 | 0.94 | 0.58–1.5 | 0.78 |
| high conidia | −0.28 | 0.33 | 0.76 | 0.40–1.45 | 0.40 |
| fungus-killed | −0.25 | 0.24 | 0.78 | 0.49–1.25 | 0.30 |
| sporulating | −3.18 | 1.00 | 0.041 | 0.0060–0.30 | 0.0016 |

binomial test: $p = 0.39$, $n = 34$; Ctrl versus HC: 50% versus 50%, binomial test: $p = 1$, $n = 14$). Quite remarkably, a significantly smaller percentage of prey-retrieving ants (only 22.6%) preferred the fly that was recently killed by the fungus in comparison with 77.4% of the ants that chose the control fly (binomial test: $p = 0.003$, $n = 31$). Likewise, ants refrained from retrieving the sporulating fly in comparison with the control prey regardless of whether the control item was a decaying fly (binomial test: $p < 0.001$, $n = 31$) or a freshly dead control fly (binomial test: $p < 0.001$, $n = 16$). Ants equally touched the two available prey with a total number of four antennal contacts for the Ctrl–Ctrl, Ctrl–LC, Ctrl–HC and Ctrl–FKill conditions and a total number of six antennal contacts for the conditions that included a sporulating prey (Ctrl–Spo and Dc–Spo conditions). Nevertheless, the contact indices i.e. the difference in the number of contacts made on the infected and the control fly, were not significantly different for all the tested conditions (Wilcoxon signed-rank test: all *p*-value > 0.05). This indicates that ant workers came into contact equally with both prey regardless of their infectiousness. However, ants spent less time touching a fungus-killed or a sporulating fly (table 3). Indeed, the contact duration indices, were significantly lower than zero in conditions where the fly recently died from fungal infection ($m = -18$ s $[-21,109]$, $n = 71$; Wilcoxon signed-rank test: $p = 0.008$;

**Table 3.** For ants that contacted both prey, duration of contacts and self-grooming are provided for the different experimental conditions. Median, first and third quartiles values are given. We marked in bold the contaminated fly in each condition and the corresponding contact duration. Since the condition Ctrl-Ctrl did not contain a contaminated fly, we arbitrarily designated one of the two prey as the contaminated one. The percentage of self-grooming ants were compared between experimental conditions using a $\chi^2$-test of homogeneity. We used a Kruskal–Wallis test to compare the self-grooming time spent by ants between experimental conditions.

| experimental conditions: | | Ctrl versus **Ctrl** | Ctrl versus **LC** | Ctrl versus **HC** | Ctrl versus **FKill** | Ctrl versus **Spo** | Dc Versus **Spo** |
|---|---|---|---|---|---|---|---|
| **contact duration** | median [Q1, Q3](s) | 52 [12,85] versus **54 [17,111]** ($n = 60$) | 52 [11,129] versus **56 [7,134]** ($n = 62$) | 32 [9,104] versus **67 [15,176]** ($n = 40$) | 62 [10,144] versus **16 [3,82]** ($n = 71$) | 96 [58,182] versus **10 [3,24]** ($n = 58$) | 122 [38,214] versus **9 [3,21]** ($n = 106$) |
| **self-grooming** | % of ants | 36.70% | 43.60% | 57.50% | 60.60% | 62.10% | 55.70% |
| | | | | Chi-squared test : $\chi^2 = 12.8$, d.f. = 5, $p = 0.025$ | | | |
| **grooming duration** | median [Q1 Q3](s) | 64 [38,80] ($n = 22$) | 51 [28,86] ($n = 27$) | 51 [35,74] ($n = 23$) | 53 [28,80] ($n = 43$) | 34 [6,54] ($n = 36$) | 52 [17,79] ($n = 59$) |
| | | | | Kruskal-Wallis test : $H = 9.5$, d.f. = 5, $p = 0.09$ | | | |

figure 6) and when the fly was sporulating (Wilcoxon signed-rank test: $p < 0.001$ for the Ctrl–Spo: $m = -85$ s [−167,−15], $n = 58$ and for the Dc–Spo condition: $m = -100$ s [−195,−24], $n = 106$).

Concerning the percentage of self-grooming ants, they significantly differed between experimental conditions ($\chi^2$-test: $\chi^2 = 12.816$, d.f. = 5, $p = 0.025$; table 3) but not in pairwise comparison (all $p$-values > 0.05). We observed that 36.7% and 43.6% of ants groomed themselves in the control condition and when one fly was covered with a low amount of conidia, respectively. This percentage increased to around 60% when the fly was covered with a high amount of conidia, recently fungus-killed or sporulating. The total time spent by ants grooming themselves was, however, not significantly different between experimental conditions (Kruskal–Wallis test, $H = 9.5$, d.f. = 5, $p = 0.09$; table 3).

Finally, we investigated whether the propensity of ants to retrieve the control prey was altered by the level of fungus infection associated with the nearby alternative prey. This analysis enabled us to check whether ants made their choice not only according to the infectiousness of the last inspected prey but also of the other prey that was concurrently offered in the foraging arena. In other words, we wondered whether the ants made a global assessment of the sanitary risks associated with all the encountered prey and accordingly altered their prey retrieval. We found that ants showed a similar probability of taking the control prey regardless of the nearby fungus-infected prey (Cox model, LRT: $\chi^2 = 6.33$, d.f. = 4, $p = 0.2$; figure 7, table 4). At most, one can see that ants were slightly more hesitating when the freshly dead fly was located next to a fly covered with a high concentration of conidia (HR = 0.46, $p = 0.08$) or next to a sporulating fly (HR = 0.62, $p = 0.17$; figure 7) than in the reference condition (Ctrl versus Ctrl). The median time elapsed before deciding to retrieve the control prey thus tended to be longer in the Ctrl–HC and Ctrl–Spo condition (186 s [80,243] and 146 s [70,221], respectively) compared to the reference condition (Ctrl versus Ctrl, 106 s [85,157]).

# 4. Discussion

We found that *M. rubra* ants can detect the potential for infection of a prey corpse and they preferentially retrieve food items that minimize sanitary risks for the whole colony. Overall, ants can identify a prey infected with the generalist entomopathogen fungus *M. brunneum*. However, their avoidance behaviour closely depends on the stage of fungus development. When being offered a prey covered with conidia, *M. rubra* foragers display the same behaviour and were as likely to retrieve this food item as a spore-free prey. Unlike several ant species that actively avoid a highly contaminated prey (around $3 \times 10^5$ spores per prey [20]), *M. rubra* workers were not repelled by a corpse covered with conidia, neither for a low amount (around 5000 spores) nor a high amount (around $10^6$ spores) of conidia covering the prey body. Furthermore, we could not detect any licking behaviour that would

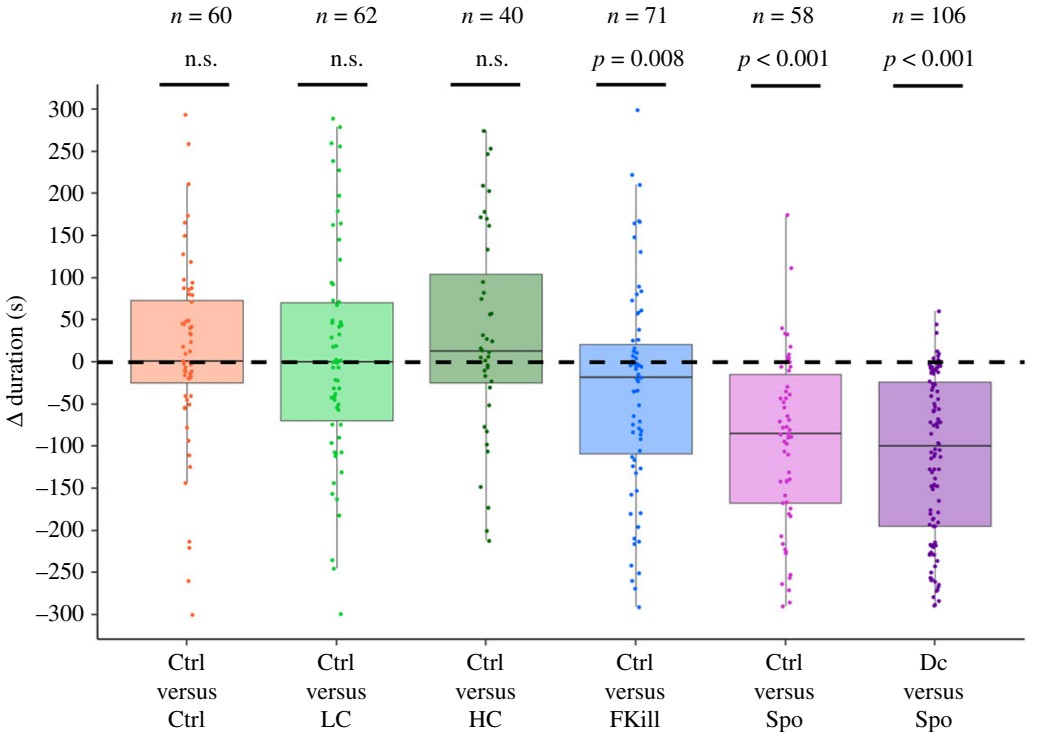

**Figure 6.** Contact duration indices (Δ duration) for each experimental condition and for the ants that contacted both prey. Indices were calculated by subtracting the total duration of contacts made on the control prey from that observed on the infected prey. Number over bar indicates sample size in each condition. We performed a Wilcoxon signed-rank test to compare the duration of contacts made by ants on prey with a theoretical value of zero.

result in the removal of spores before prey retrieval, as reported in leaf-cutting ants, with hitchhikers riding on the leaf fragments that clean them from pathogens before they reach the nest [29,30]. Further studies are, however, needed to investigate whether such a cleaning of spore-covered prey by *M. rubra* workers might take place at nest entrances or inside the nest, thus making safer the feeding on prey by nestmates or brood.

Even though *M. rubra* foragers were more hesitating, they eventually decided to retrieve an insect whose corpse was artificially covered with a number of conidia similar to the amount of spores released by a sporulating fly (H Pereira 2018 personal observations). Such a lack of avoidance of conidia-bearing food suggests that *M. rubra* foragers might fail to detect spores once they were no longer attached to a sporulating insect. A failure to detect spores or a lack of avoidance of fungal spores was also found in other contexts such as the choice of a new nesting site during colony relocation [23], or the choice of digging substrate during nest building [24]. Inside the nest, the fate of spore-covered prey—i.e. their ingestion by ant workers—may also dramatically decrease the actual risks of infection by spores. Indeed, in addition to their inactivation by the ants' immune defences, the ingested conidia have a low probability to germinate in the insect gut due to the presence of antifungal compounds or microbial symbionts [39]. For example, in the cricket *Shistocerca gregaria*, antifungal phenols are produced by the gut bacterial flora which inhibit the germination of *M. anisopliae* conidia [40]. Furthermore, the possible contamination of foragers' bodies, which could carry spores after having touched an infected item is likely to be limited. A set of well-known social behaviours such as allogrooming can reduce the pathogen load of incoming foragers and may thus prevent further exposure of nestmates to spores inside the ant colony [7]. It should also be noted that, at a low concentration of spores, the fact that the ants do not actively avoid contaminated prey could even be beneficial for the colony. Nestmates are then exposed to a pathogen load that is too small to induce a lethal infection [41] but sufficient to promote a response of the immune system. This process, known as social immunization [42], could be beneficial for the colony by increasing its resistance when being later exposed to the same pathogens [41,43].

We also found that *M. rubra* foragers retrieve a decaying prey and a freshly dead one at equal rates, as it might be expected from a scavenging ant species which regularly feed on insect corpses which have been

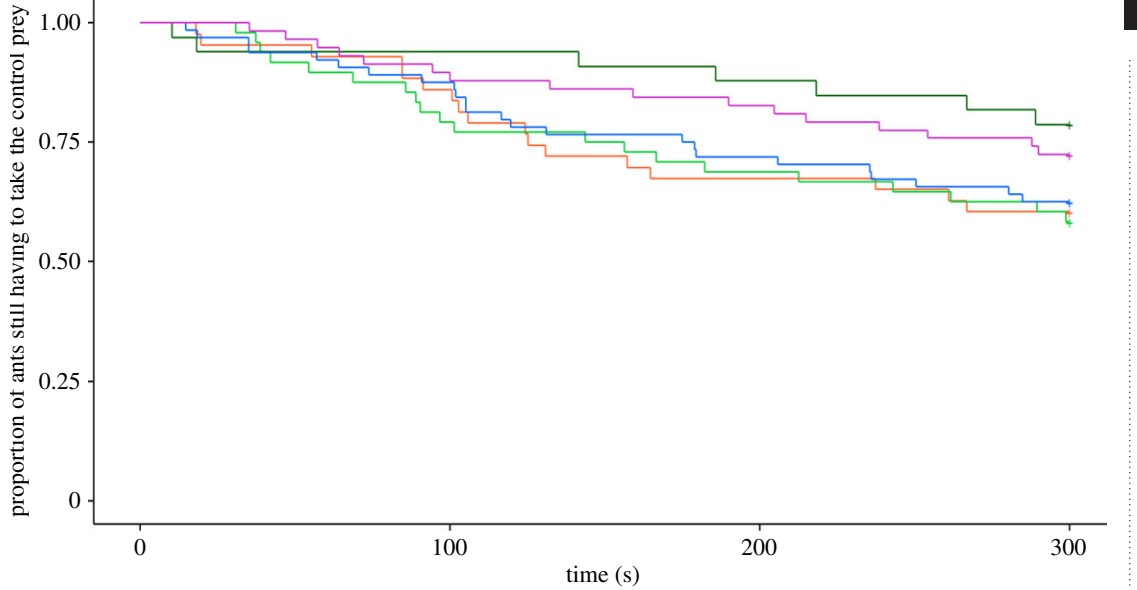

**Figure 7.** Proportion of ants that had not yet retrieved the control prey as a function of time. The percentages are given for the ant having contacted both prey for the different experimental conditions. Orange: Ctrl versus Ctrl ($n = 43$), green: Ctrl versus LC ($n = 48$), dark green: Ctrl versus HC ($n = 33$), blue: Ctrl versus FKill ($n = 64$), pink: Ctrl versus Spo ($n = 58$).

**Table 4.** Effect of infectiousness of nearby prey on the time elapsed to choose the control one. Values of estimate parameter, standard error (s.e.), hazard ratio (HR) and confidence interval (CI) of the Cox proportional hazard regression models are given. $p$-Values compare the likelihood of a taking event between a given condition and the reference condition (i.e. Ctrl versus Ctrl).

| variable (reference) | estimate | ±s.e. | HR | 95% CI | $p$-value |
|---|---|---|---|---|---|
| Ctrl versus LC | 0.05 | 0.33 | 1.06 | 0.55–2.02 | 0.87 |
| Ctrl versus HC | −0.78 | 0.45 | 0.46 | 0.19–1.10 | 0.08 |
| Ctrl versus FKill | −0.08 | 0.32 | 0.92 | 0.50–1.72 | 0.80 |
| Ctrl versus Spo | 0.62 | 0.35 | 0.62 | 0.31–1.23 | 0.17 |

dead for several days. Further studies are needed to investigate whether scavenging ants benefit from enhanced digestive and immune defences against pathogens associated with decaying insects. Quite remarkably, the foraging strategy of ants differed once the decaying prey started to sporulate. As soon as a high risk of infection originated from a sporulating cadaver, foragers showed an active, distinct avoidance behaviour. Our results are in line with Marikovsky's field study showing that *Formica rufa* ants never feed on sporulating nestmates infected by *Tarichium* fungi [44]. Such an avoidance behaviour may explain why sporulating insects seem to remain in the environment for weeks while uninfected decaying prey are quickly eaten by a wide range of scavengers like birds or other invertebrates [45]. Put together, these results suggest that sporulating corpses are unattractive and even repellent to scavenging ants that refrain from retrieving such food items. The volatile organic compounds (VOCs) released either by virulent strains of *M. anisopliae* conidia [25] or by fungus grown on agar media [46] also trigger avoidance behaviour in termites [25]. For example, *M. anisopliae* fungus emits VOCs such as furanone, 2-phenylpropenal, 2,5,5-trimethyl-1-hexene, *n*-tetradecane and 2,6-dimethylheptadecane of which the relative amount varies depending on the virulence of the strain [47]. However, it is worth noting that *M. rubra* workers behaved differently when conidia originated from a sporulating fly or when spores were topically applied onto the prey's body. This suggests that the clear-cut avoidance of sporulating corpses is triggered not only by physical contacts with fungal spores or by spore-specific VOCs but also by the detection of VOCs released by external structures of the sporulating fungus [47]. To our knowledge, no study has examined the differences between the VOC profiles of fungal conidia and those of fungus originating from a sporulating insect cadaver. Quantitative and/or qualitative differences in the released

VOCs could explain the differential avoidance behaviour of ants to these two stages of fungal development. An alternative non-exclusive explanation for the ants' reluctance to take a sporulating prey could be that they no longer perceive this corpse as a food item but rather as waste. Since fungal mycelium has colonized the whole corpse, the sporulating fly may no longer provide valuable nutrients to foragers that disregard such food resources.

Finally, our study strongly suggests that the population of M. rubra foragers is not homogeneous, with individuals differing in their sensitivity to sanitary risks. Most workers avoid retrieving a sporulating insect but only some of them were able to detect a prey recently died from fungal infection that did not yet show any signs of sporulation. For the individuals that immediately retrieved the first prey encountered, they took the fly regardless of whether it was killed by an entomopathogenic fungus or by exposure to cold. On the other hand, for the foragers that had contact with both prey, some individuals preferred to retrieve the uncontaminated control than the fungus-killed prey. In other words, when able to compare the two prey, some individuals can discriminate the cause of their death. The early discrimination of infected but not yet sporulating food was already reported in subterranean termites and in other insects that did not cannibalize cadavers once they were killed by entomopathogenic fungi (M. anisopliae [48,49]). This detection may rely on pathogen-induced changes in the cuticular profile of prey. As for bees parasitized by Varroa jacobsoni [50] or by bacteria [51], fungus-induced changes in the chemical composition of the host cuticle may take place during post-mortem hours [52] while its hyphae grow inside the host body and degrade the cuticle to begin its external phase of growth [53,54]. We assume that short-term changes in the chemical profile of prey can allow the early detection of the fungus and avoidance behaviour among some of the ant foragers. Further studies are needed to characterize these individuals, namely with respect to their perceptive abilities, their immunity, their age or their personal experience.

From the colony point of view, avoiding fungus-infected food is a first line of defence against sanitary risks [55]. However, when an exploring ant fails to detect a contaminated food and takes it back to the nest, some additional filters may limit the sanitary threat for the whole colony, for example potentially harmful food items might be discarded by other workers wandering in the nest surroundings as well as by gatekeepers. Through this multiple assessment, contaminated food may never reach the nest interior nor be delivered to consumers. For decaying prey, which host a potentially pathogenic microfauna as well for conidia-covered or fungus-killed prey that are nevertheless retrieved by foragers, the main 'sanitary filters' could be located downstream in the food distribution process, in particular by being set apart from the queen or larvae. Furthermore, as for the carnivorous ant species, M. rubra, F. rufa and Polyrhachis dives [20], scavenging ants could limit sanitary risks by not storing prey inside the nest but instead, by consuming this type of food items in the next following days. Further studies should explore the set of behavioural, physiological and socio-spatial mechanisms of pathogen avoidance that limit sanitary risks once a potentially harmful prey is brought inside the ant colony. Beyond the foraging context, strategies of pathogen avoidance are highly variable and may lead to contrasted responses to a fungal threat. While some ant species avoid walking on contaminated areas [20], others are attracted to fungus-infected nesting places [23] or equally dig their nest in contaminated soil [24]. This suggests that the attraction/avoidance response to pathogens is a complex behaviour that is ruled not only by the strain virulence and the pathogens' prevalence, but also by the context of exposure and the life history of the insect host.

Ethics. No licences or permits were required for this research. Ant colonies were collected with care in the field and were maintained in nearly natural conditions in the laboratory. Ants were provided with suitable nesting sites, food and water, thus minimizing any adverse impact on their welfare. After the experiments, the rest of the colony was kept in the laboratory and reared until their natural death.

Data accessibility. Data used in this article are available in the Zenodo Digital Repository: https://zenodo.org/record/3462814#.XY1LYEYzZEY.

Authors' contributions. C.D. and H.P. participated in the design of the study. H.P. acquired the data. H.P. and C.D. carried out data analysis and statistical analyses. C.D. and H.P. wrote the manuscript. All the authors gave their final approval for publication.

Competing interests. The authors have no competing interests that might be perceived to influence the results and/or discussion reported in this paper.

Funding. H.P. was financially supported by a PhD grant from the FRIA (Fonds pour la formation à la Recherche dans l'Industrie et dans l'Agriculture). This research was funded by a research credit (grant nos. CDR J.0092.16 and CDR J.0053.18F) from FRS-FNRS. C.D. is Research Director from the Belgian National Fund for Scientific Research (FNRS).

Acknowledgements. We wanted to thanks Léo Suret who participated in some of the experiments. We also thank Nell Foster for proofreading the manuscript.

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
