## [Reviewer comments · Royal Society Open Science]

Review History

RSOS-191705.R0 (Original submission)

Review form: Reviewer 1

Is the manuscript scientifically sound in its present form?

No

Are the interpretations and conclusions justified by the results?

Yes

Is the language acceptable?

Yes

Do you have any ethical concerns with this paper?

No

Have you any concerns about statistical analyses in this paper?

Yes

Recommendation?

Major revision is needed (please make suggestions in comments)

Comments to the Author(s)

The present study investigates whether ants can and do distinguish pathogen infected prey in different stages of pathogen development. Binary choice experiment tested six different prey conditions, starting with uninfected prey as a control item and ended with sporulating prey after it was artificially infected with *Metarhizium brunneum*. The results show that prey with a high pathogenic infection risk were avoided while prey with low pathogenic risk was taken to be fed to the colony.

The experiments are well designed and sample sizes for each test are high, the ensuing results are interesting. However, my concern regarding the results and the conclusions lies in the colonies used to perform the study and the eligibility of the performed statistical test.

As was stated, one of the experimental series (Control vs. prey with high amount of dry conidia (HC)) was performed one year later than the rest and with entirely different colonies. These might have been sampled in the same region as the former used colonies, which could indicate a high level of relatedness between colonies, but there is still the possibility, that, for example the different time of colony collection (fall 2017 vs. spring 2019) could affect the ants behaviour. Some of the results obtained with these colonies seem to stick out of the observed pattern, at least when looking at some graphs. I understand that sometimes due to circumstances some experiments will be performed later than the rest of the study, which might also involve the testing of new animals. However, this should then in some way be incorporated into the statistical tests performed. Unless the authors can give a strong reasoning about treating and comparing these results statistically the same, I have reservations about rendering the study and the outlook solid. I would therefore suggest to analyse the results using a model (GLM or GLMM) that includes 'Colony' as a factor. This would give clear testament as to whether the four different colonies used influenced the observed behaviours.

I also have a number of more detailed comments I outline below.

Abstract

Page 2, line 39: You write, that there were six established conditions, and then name only 4. Please include the other 2. Without this information it was rather confusing to follow the results summarized in the latter sentences in the abstract.

As I understand from the abstract, the authors artificially subjected prey items to the entomopathogen fungus, to which the ants did not react with avoidance of the prey items. Latter they write that the ants avoided sporulating flies and freshly dead flies were caught less. Does this mean that the non-avoidance of prey items was just for dry conidia (be it low or high concentration)? I would consider rewriting a bit to emphasize that while all infection was artificial, artificial covering means dry conidia.

Page 2, line 42: How were the flies naturally infected with *Metarhizium*?

Page 2, line 44-45: Do the ants really not detect dry conidia presence or are they, in this instance capable, to remove dry conidia by cleaning the item and so can make use of the prey to feed the colony? Leaf-cutting ants, by hitchhikers riding on the leaves back to the nest, clean their leaf-fragments of pathogens to reduce pathogen load before they reach the nest.

Introduction

Page 4, line 67: You write that evidence for parasite pressure is lacking, because they have developed effective defences at the group level. I would like to disagree with the authors' reasoning here and argue, that the development of defence mechanisms, for example grooming behaviour or the production of antipathogenic substances in the ants glands is evidence for the pressure that parasites or pathogens put on the members of the group. Please rewrite.

Page 5, line 115: As this paragraph outlines avoidance behaviour of social insects to compromised food sources, I would like to suggest also these references to include, as they relate directly to the topic of the manuscript, both showing research of 'food' avoidance as well as food cleaning behaviour in leaf-cutting ants foraging on life plant matter.

Coblentz & Van Bael 2013, Field colonies of leaf-cutting ants select plant materials containing low abundances of endophytic fungi. *Ecosphere*, 4, Article 66

Griffith and Hughes 2010, Hitchhiking and the removal of microbial contaminants by the leaf-cutting ant *Atta colombica*, *Ecological Entomology* 35, 529-537

Vieira-Neto et al. 2006. Hitch-hiking behaviour in leaf-cutter ants: an experimental evaluation of three hypotheses. *Insectes Sociaux*, 53, 326-332

Page 5, line 125: ...shared with individuals

Page 5, line 136: hemocoel

Material and Methods

Page 5, line 147: Were these colonies queenright or queenless? Was any brood collected? When were the colonies collected?

Could you also briefly outline the life history of this species (i.e. colony size, nest site, distribution, etc.)

Page 6, line 169-170: 'leads the insect to death' - please rewrite. 'causes the death of the insect' for example.

Page 6, 180-181: This means, that after 2 days you acquired a sporulating corpse, I assume. Freshly dead corpses were washed as described but then immediately used for testing and not put into Petri dishes? For clarity, please state very clearly at which step you would get a freshly dead corpse and when the result was a sporulating one.

Page 7, line 189: homogenised

Page 7, line 201: Please give this information at the beginning of the Materials and Methods section (as see my question above).

Page 7, line 203: of food

Page 7, line 206: consisted of

Page 7, line 215: Do you have any indication how high or low these control flies are in their pathogen load?

Page 7, line 231-233: Please see my question above (page 6, 180-181) in this regard and include this information earlier in the M&M section.

Page 8, line 243: Here you write that you tested 136 ants under control prey - sporulating prey (Ctrl - Spo) conditions. In the supplementary material, Table S.1 the total number of ants tested under these conditions (column 6) is 75. Vica versa in line 249 you give 75 as sample size when testing decaying prey vs sporulating prey and in Table S.1 the sample size is given as 136. Please check and change accordingly either in the main body of the text or in the supplementary material.

Page 8, line 245: When reading I was a bit unclear about the procedure for the decaying prey. Did it go through a sham treatment of conidia (only solvent but no actual conidia), then killed by cold, kept in a petri dish at 25°C for 2 days and then offered as prey? It would be less confusing to just give in a short sentence the information about the whole treatment about how you acquired decayed prey.

Page 8, line 250: except

Page 8, line 251: I find it a bit worrying that one of the experimental series was performed with other colonies than the other 5 ones. At least you should include a justification why these results can be used in the statistical analysis together with the other 5 experimental series which were all performed with the same colonies.

Page 8, line 255: How was this contact achieved? Antenna, legs or mouthparts?

Page 8, line 272: I believe there is a word missing at the end of the sentence?

Page 8, line 272: performed Dunn's post-hoc tests

Page 8, line 273: using the "FSA" package

Results

Page 9, line 305: See my earlier comments above about treating the series HC, that was performed using entirely different colonies, as if it were the same as the other 5 performed series. Either clearly justify your choice of using these results and why the use of the chosen statistics is still appropriate, or, as I would strongly suggest, use a statistical model (like GLM or GLMM) that incorporates 'colony' as a factor. Then you would clearly know whether using different colonies led to the drop in prey retrieval (or other results) even below the percentage of flies killed by fungal infection.

Page 9, line 310: Please give the actual percentages for conditions Ctrl vs. Spo and Dc vs. Spo rather than 'around 35%'.

Page 9, line 314: one of the two prey

Page 9, line 315: How did the ants behave that only chose one prey? Did they walk on a direct path towards the one prey, did they find it after walking around the arena? Please describe a bit the walking/approaching behaviour of the ant towards the fly.

Page 10, line 329: except

Page 10, line 336: associated with

Page 10, line 354: What about the decaying flies (Dc)? So far you mentioned Ctrl, LC, HC and Spo. How often was decaying prey contacted?

Page 10, line 355: please put the information regarding grooming behaviour of ants into a separate paragraph.

Page 10, line 356: depending on

Page 11, line 374: Please use another word than 'population' for the individuals contacting both prey items.

Page 11, line 382: In regards to the ants that made a choice

Page 11, line 405: See above, please use a separate paragraph for grooming behaviour.

Page 12, line 424: As you chose to emphasize these two series in regards to retrieval time of non-infected prey when infected prey was nearby, what was the actual elapsed time until retrieval in these two conditions in comparison to other conditions?

Discussion

Page 12, line 428: a prey corpse

Page 12, line 441: One could draw this conclusion as a possible explanation as you already formulated with a 'suggests' rather than a 'shows', however, it is also possible that through grooming behaviour, *M. rubra* manages to remove fungal spores and make use of the discovered prey item. While an increase in grooming was not observed in the feeding arena of the experiment, more extensive prey grooming might take place once the prey reaches the nest where it might be effectively cleaned by a group of ants. Is the lack of increased grooming in the experimental arena your only reason why you think the worker could not detect the spore load?

Page 12, line 443: such as the choice

Page 12, line 444: For reference [25], did the authors there not also argue, that excavating a nest in contaminated soil could also be a way to build up a resistance to the encountered, often harmful microorganisms?

Page 12, lines 444-457: In relation with my comment above, does this section of your discussion not rather explain, that ants might still detect the artificially applied conidia load on dead prey, but because of the described findings and behaviours, the colony can still make use of this type of prey without a high danger of infecting colony members?

Page 13, line 467: a wide range

Figures

Page 19, Figure 3: What is the reason that you changed the presentation of data in comparison with Figure 2. In both cases you present percentage data of workers performing a task (either prey retrieval or contacting one prey item) or not performing/performing another task (no prey retrieval or contacting both prey items). Both tasks together amount to 100%, composed of grey bar on top of a black bar. In Figure 3, the grey bar is now set to the right instead of on top of the black bar.

The data can be displayed either way, I was just wondering whether there was any significance to it? If not, I would think about sticking to one type of figure.

Page 19, line 672-673: 'Control regroups all prey used as control in choice tests.' I do not understand this sentence.

Page 20, line 686: ants retrieving the control prey (or the decaying prey in the last series).

Page 22, line 698: retrieved

Page 22, line 700: Red: Ctrl vs Ctrl. You should also either use abbreviations for every series, as you did for the first three series or the formulated words as you did for the last two series.

Bibliography

Please take care to put all the species names in italics. Also, edit for citation style, don't use a mix of either starting with a major letter and the rest in minor, all nouns start with a major letter, all words are written in major letters. I suppose this is the result of using Mendeley/Zotero/Endnote where this has not been edited.

Review form: Reviewer 2

Is the manuscript scientifically sound in its present form?

Yes

Are the interpretations and conclusions justified by the results?

Yes

Is the language acceptable?

Yes

Do you have any ethical concerns with this paper?

No

Have you any concerns about statistical analyses in this paper?

No

Recommendation?

Accept with minor revision (please list in comments)

Comments to the Author(s)

The authors tested *Myrmica* ants for avoidance behavior towards fly prey at different stages of fungal pathogen infection. The authors show that the ants do not discriminate against prey covered in conidia, but chose healthy over infected prey at later stages of infection, particularly at the stage of sporulation, when the ants also perform more selfgrooming after prey handling. The experiments are well-designed, the data clearly presented and discussed. The paper reads very well, except for the abstract .

- 1) In the abstract, it is unclear what the authors mean with artificial vs natural infection (in fact, both are artificial, but the first is covering a healthy prey in fungal conidia, vs the other is a later stage of disease). "prey" is missing after "sporulating". Also, the last half sentence is highly speculative and has not been addressed in the manuscript and hence should be removed
- 2) The references from 34 onwards are all shifted by 2, which needs to be corrected.
- 3) Technically, topical application of conidia by a Tween or Triton X (were really both used?) should not be referred to as "dry spores" (l 183), which would reflect application of dry conidia from agar plate in the absence of a surfactant.
- 4) l 272 sentence incomplete

Review form: Reviewer 3

Is the manuscript scientifically sound in its present form?

Yes

Are the interpretations and conclusions justified by the results?

Yes

Is the language acceptable?

Yes

Do you have any ethical concerns with this paper?

No

Have you any concerns about statistical analyses in this paper?

No

Recommendation?

Accept as is

Comments to the Author(s)

This manuscript represents a comprehensive collection of behavioral experiments in which *Myrmica rubra* ants were allowed to forage for *Drosophila* cadavers that were recently killed, decaying, covered in fungal spores of *Metarhizium brunneum* or killed by infection with this fungal species. The authors performed choice experiments in which ants showed a significant difference between interaction with and collection of sporulating cadavers and all other offered fly types. The data was analyzed using appropriate statistical methods and accompanied by both a comprehensive introduction and discussion. This makes me conclude that this work is acceptable for publication as is.

My only suggestion to the authors is that they might want to look at their data for the individual ants and how they were showing either more or less risky behavior with regards to the prey that they were presented with. The authors found individual variation, which is very interesting. This makes me wonder if the potential scouts were more risk-taking (aka more exploratory) while the regular foragers might have been less so. Is there a way for the authors to backtrack if perhaps the first ants entering the foraging arena (the potential scouts) were introducing the seen variation in individual behavior versus ants that might have shown up later to forage for prey?

Furthermore, they might want to look at the following lines and consider re-writing them for clarity or checking them for perceived typos:

Line 91: consider revising "fungi whose some strains"

Lines 96-99: very long and difficult to follow sentence. Consider revising.

Line 306: "fly" probably needs to be "flies"

Line 314: add the word "of" to the end of the line to "one of the two prey"

Decision letter (RSOS-191705.R0)

14-Nov-2019

Dear Mr Pereira,

The editors assigned to your paper ("Pathogen avoidance and prey discrimination in ants") have now received comments from reviewers. We would like you to revise your paper in accordance with the referee and Associate Editor suggestions which can be found below (not including confidential reports to the Editor). Please note this decision does not guarantee eventual acceptance.

Please submit a copy of your revised paper before 07-Dec-2019. Please note that the revision deadline will expire at 00.00am on this date. If we do not hear from you within this time then it will be assumed that the paper has been withdrawn. In exceptional circumstances, extensions

may be possible if agreed with the Editorial Office in advance. We do not allow multiple rounds of revision so we urge you to make every effort to fully address all of the comments at this stage. If deemed necessary by the Editors, your manuscript will be sent back to one or more of the original reviewers for assessment. If the original reviewers are not available, we may invite new reviewers.

- Data accessibility

If you wish to submit your supporting data or code to Dryad (<http://datadryad.org/>), or modify your current submission to dryad, please use the following link:
<http://datadryad.org/submit?journalID=RSOS&manu=RSOS-191705>

- Competing interests

- Authors' contributions

- Acknowledgements

- Funding statement

Kind regards,
Lianne Parkhouse
Editorial Coordinator
Royal Society Open Science
openscience@royalsociety.org

on behalf of the Associate Editor, and Professor Kevin Padian (Subject Editor)
openscience@royalsociety.org

Associate Editor's comments to the Author:

Please ensure you respond fully to the reviewer concerns, and provide a point-by-point response to them (as well as a tracked-changes version of the paper to aid the editors, and reviewers identifying the changes). The revision may need further review, but the general view of the manuscript from the reviewers appears positive.

Reviewers' Comments to Author:

Reviewer: 1
Comments to the Author(s)

The present study investigates whether ants can and do distinguish pathogen infected prey in different stages of pathogen development. Binary choice experiment tested six different prey conditions, starting with uninfected prey as a control item and ended with sporulating prey after it was artificially infected with *Metarhizium brunneum*. The results show that prey with a high pathogenic infection risk were avoided while prey with low pathogenic risk was taken to be fed to the colony.

The experiments are well designed and sample sizes for each test are high, the ensuing results are interesting. However, my concern regarding the results and the conclusions lies in the colonies used to perform the study and the eligibility of the performed statistical test.

As was stated, one of the experimental series (Control vs. prey with high amount of dry conidia (HC)) was performed one year later than the rest and with entirely different colonies. These might have been sampled in the same region as the former used colonies, which could indicate a high level of relatedness between colonies, but there is still the possibility, that, for example the different time of colony collection (fall 2017 vs. spring 2019) could affect the ants behaviour. Some

of the results obtained with these colonies seem to stick out of the observed pattern, at least when looking at some graphs. I understand that sometimes due to circumstances some experiments will be performed later than the rest of the study, which might also involve the testing of new animals. However, this should then in some way be incorporated into the statistical tests performed. Unless the authors can give a strong reasoning about treating and comparing these results statistically the same, I have reservations about rendering the study and the outlook solid. I would therefore suggest to analyse the results using a model (GLM or GLMM) that includes 'Colony' as a factor. This would give clear testament as to whether the four different colonies used influenced the observed behaviours.

I also have a number of more detailed comments I outline below.

Abstract

Page 2, line 39: You write, that there were six established conditions, and then name only 4. Please include the other 2. Without this information it was rather confusing to follow the results summarized in the latter sentences in the abstract.

As I understand from the abstract, the authors artificially subjected prey items to the entomopathogen fungus, to which the ants did not react with avoidance of the prey items. Later they write that the ants avoided sporulating flies and freshly dead flies were caught less. Does this mean that the non-avoidance of prey items was just for dry conidia (be it low or high concentration)? I would consider rewriting a bit to emphasize that while all infection was artificial, artificial covering means dry conidia.

Page 2, line 42: How were the flies naturally infected with *Metarhizium*?

Page 2, line 44-45: Do the ants really not detect dry conidia presence or are they, in this instance capable, to remove dry conidia by cleaning the item and so can make use of the prey to feed the colony? Leaf-cutting ants, by hitchhikers riding on the leaves back to the nest, clean their leaf-fragments of pathogens to reduce pathogen load before they reach the nest.

Introduction

Page 4, line 67: You write that evidence for parasite pressure is lacking, because they have developed effective defences at the group level. I would like to disagree with the authors' reasoning here and argue, that the development of defence mechanisms, for example grooming behaviour or the production of antipathogenic substances in the ants glands is evidence for the pressure that parasites or pathogens put on the members of the group. Please rewrite.

Page 5, line 115: As this paragraph outlines avoidance behaviour of social insects to compromised food sources, I would like to suggest also these references to include, as they relate directly to the topic of the manuscript, both showing research of 'food' avoidance as well as food cleaning behaviour in leaf-cutting ants foraging on life plant matter.

Coblentz & Van Bael 2013, Field colonies of leaf-cutting ants select plant materials containing low abundances of endophytic fungi. *Ecosphere*, 4, Article 66

Griffith and Hughes 2010, Hitchhiking and the removal of microbial contaminants by the leaf-cutting ant *Atta colombica*, *Ecological Entomology* 35, 529-537

Vieira-Neto et al. 2006. Hitch-hiking behaviour in leaf-cutter ants: an experimental evaluation of three hypotheses. *Insectes Sociaux*, 53, 326-332

Page 5, line 125: ...shared with individuals

Page 5, line 136: hemocoel

Material and Methods

Page 5, line 147: Were these colonies queenright or queenless? Was any brood collected? When were the colonies collected?

Could you also briefly outline the life history of this species (i.e. colony size, nest site, distribution, etc.)

Page 6, line 169-170: 'leads the insect to death' – please rewrite. 'causes the death of the insect' for example.

Page 6, 180-181: This means, that after 2 days you acquired a sporulating corpse, I assume. Freshly dead corpses were washed as described but then immediately used for testing and not put into Petri dishes? For clarity, please state very clearly at which step you would get a freshly dead corpse and when the result was a sporulating one.

Page 7, line 189: homogenised

Page 7, line 201: Please give this information at the beginning of the Materials and Methods section (as see my question above).

Page 7, line 203: of food

Page 7, line 206: consisted of

Page 7, line 215: Do you have any indication how high or low these control flies are in their pathogen load?

Page 7, line 231-233: Please see my question above (page 6, 180-181) in this regard and include this information earlier in the M&M section.

Page 8, line 243: Here you write that you tested 136 ants under control prey – sporulating prey (Ctrl – Spo) conditions. In the supplementary material, Table S.1 the total number of ants tested under these conditions (column 6) is 75. Vica versa in line 249 you give 75 as sample size when testing decaying prey vs sporulating prey and in Table S.1 the sample size is given as 136. Please check and change accordingly either in the main body of the text or in the supplementary material.

Page 8, line 245: When reading I was a bit unclear about the procedure for the decaying prey. Did it go through a sham treatment of conidia (only solvent but no actual conidia), then killed by cold, kept in a petri dish at 25°C for 2 days and then offered as prey? It would be less confusing to just give in a short sentence the information about the whole treatment about how you acquired decayed prey.

Page 8, line 250: except

Page 8, line 251: I find it a bit worrying that one of the experimental series was performed with other colonies than the other 5 ones. At least you should include a justification why these results can be used in the statistical analysis together with the other 5 experimental series which were all performed with the same colonies.

Page 8, line 255: How was this contact achieved? Antenna, legs or mouthparts?

Page 8, line 272: I believe there is a word missing at the end of the sentence?

Page 8, line 272: performed Dunn's post-hoc tests

Page 8, line 273: using the "FSA" package

Results

Page 9, line 305: See my earlier comments above about treating the series HC, that was performed using entirely different colonies, as if it were the same as the other 5 performed series. Either clearly justify your choice of using these results and why the use of the chosen statistics is still appropriate, or, as I would strongly suggest, use a statistical model (like GLM or GLMM) that incorporates 'colony' as a factor. Then you would clearly know whether using different colonies led to the drop in prey retrieval (or other results) even below the percentage of flies killed by fungal infection.

Page 9, line 310: Please give the actual percentages for conditions Ctrl vs. Spo and Dc vs. Spo rather than 'around 35%'.

Page 9, line 314: one of the two prey

Page 9, line 315: How did the ants behave that only chose one prey? Did they walk on a direct path towards the one prey, did they find it after walking around the arena? Please describe a bit the walking/approaching behaviour of the ant towards the fly.

Page 10, line 329: except

Page 10, line 336: associated with

Page 10, line 354: What about the decaying flies (Dc)? So far you mentioned Ctrl, LC, HC and Spo. How often was decaying prey contacted?

Page 10, line 355: please put the information regarding grooming behaviour of ants into a separate paragraph.

Page 10, line 356: depending on

Page 11, line 374: Please use another word than 'population' for the individuals contacting both prey items.

Page 11, line 382: In regards to the ants that made a choice

Page 11, line 405: See above, please use a separate paragraph for grooming behaviour.

Page 12, line 424: As you chose to emphasize these two series in regards to retrieval time of non-infected prey when infected prey was nearby, what was the actual elapsed time until retrieval in these two conditions in comparison to other conditions?

Discussion

Page 12, line 428: a prey corpse

Page 12, line 441: One could draw this conclusion as a possible explanation as you already formulated with a 'suggests' rather than a 'shows', however, it is also possible that through grooming behaviour, *M. rubra* manages to remove fungal spores and make use of the discovered

prey item. While an increase in grooming was not observed in the feeding arena of the experiment, more extensive prey grooming might take place once the prey reaches the nest where it might be effectively cleaned by a group of ants. Is the lack of increased grooming in the experimental arena your only reason why you think the worker could not detect the spore load?

Page 12, line 443: such as the choice

Page 12, line 444: For reference [25], did the authors there not also argue, that excavating a nest in contaminated soil could also be a way to build up a resistance to the encountered, often harmful microorganisms?

Page 12, lines 444-457: In relation with my comment above, does this section of your discussion not rather explain, that ants might still detect the artificially applied conidia load on dead prey, but because of the described findings and behaviours, the colony can still make use of this type of prey without a high danger of infecting colony members?

Page 13, line 467: a wide range

Figures

Page 19, Figure 3: What is the reason that you changed the presentation of data in comparison with Figure 2. In both cases you present percentage data of workers performing a task (either prey retrieval or contacting one prey item) or not performing/performing another task (no prey retrieval or contacting both prey items). Both tasks together amount to 100%, composed of grey bar on top of a black bar. In Figure 3, the grey bar is now set to the right instead of on top of the black bar.

The data can be displayed either way, I was just wondering whether there was any significance to it? If not, I would think about sticking to one type of figure.

Page 19, line 672-673: 'Control regroupes all prey used as control in choice tests.' I do not understand this sentence.

Page 20, line 686: ants retrieving the control prey (or the decaying prey in the last series).

Page 22, line 698: retrieved

Page 22, line 700: Red: Ctrl vs Ctrl. You should also either use abbreviations for every series, as you did for the first three series or the formulated words as you did for the last two series.

Bibliography

Please take care to put all the species names in italics. Also, edit for citation style, don't use a mix of either starting with a major letter and the rest in minor, all nouns start with a major letter, all words are written in major letters. I suppose this is the result of using Mendeley/Zotero/Endnote where this has not been edited.

Reviewer: 2

Comments to the Author(s)

The authors tested *Myrmica* ants for avoidance behavior towards fly prey at different stages of fungal pathogen infection. The authors show that the ants do not discriminate against prey covered in conidia, but chose healthy over infected prey at later stages of infection, particularly at the stage of sporulation, when the ants also perform more selfgrooming after prey handling. The experiments are well-designed, the data clearly presented and discussed. The paper reads very well, except for the abstract .

- 1) In the abstract, it is unclear what the authors mean with artificial vs natural infection (in fact, both are artificial, but the first is covering a healthy prey in fungal conidia, vs the other is a later stage of disease). "prey" is missing after "sporulating". Also, the last half sentence is highly speculative and has not been addressed in the manuscript and hence should be removed
- 2) The references from 34 onwards are all shifted by 2, which needs to be corrected.
- 3) Technically, topical application of conidia by a Tween or Triton X (were really both used?) should not be referred to as "dry spores" (l 183), which would reflect application of dry conidia from agar plate in the absence of a surfactant.
- 4) l 272 sentence incomplete

Reviewer: 3

Comments to the Author(s)

This manuscript represents a comprehensive collection of behavioral experiments in which *Myrmica rubra* ants were allowed to forage for *Drosophila* cadavers that were recently killed, decaying, covered in fungal spores of *Metarhizium brunneum* or killed by infection with this fungal species. The authors performed choice experiments in which ants showed a significant difference between interaction with and collection of sporulating cadavers and all other offered fly types. The data was analyzed using appropriate statistical methods and accompanied by both a comprehensive introduction and discussion. This makes me conclude that this work is acceptable for publication as is.

My only suggestion to the authors is that they might want to look at their data for the individual ants and how they were showing either more or less risky behavior with regards to the prey that they were presented with. The authors found individual variation, which is very interesting. This makes me wonder if the potential scouts were more risk-taking (aka more exploratory) while the regular foragers might have been less so. Is there a way for the authors to backtrack if perhaps the first ants entering the foraging arena (the potential scouts) were introducing the seen variation in individual behavior versus ants that might have shown up later to forage for prey?

Furthermore, they might want to look at the following lines and consider re-writing them for clarity or checking them for perceived typos:

Line 91: consider revising "fungi whose some strains"

Lines 96-99: very long and difficult to follow sentence. Consider revising.

Line 306: "fly" probably needs to be "flies"

Line 314: add the word "of" to the end of the line to "one of the two prey"

Author's Response to Decision Letter for (RSOS-191705.R0)

See Appendix A.

RSOS-191705.R1 (Revision)

Review form: Reviewer 1

Is the manuscript scientifically sound in its present form?

Yes

Are the interpretations and conclusions justified by the results?

Yes

Is the language acceptable?

Yes

Do you have any ethical concerns with this paper?

No

Have you any concerns about statistical analyses in this paper?

No

Recommendation?

Accept as is

Comments to the Author(s)

The authors put a lot of effort into improving the manuscript, it was a pleasure to read it. This work expands the knowledge about another aspect of social insects social immunity. I have no longer any reservation about its publication.

Review form: Reviewer 2

Is the manuscript scientifically sound in its present form?

Yes

Are the interpretations and conclusions justified by the results?

Yes

Is the language acceptable?

Yes

Do you have any ethical concerns with this paper?

No

Have you any concerns about statistical analyses in this paper?

No

Recommendation?

Accept as is

Comments to the Author(s)

The authors performed a careful revision.

Decision letter (RSOS-191705.R1)

28-Jan-2020

Dear Mr Pereira,

It is a pleasure to accept your manuscript entitled "Pathogen avoidance and prey discrimination

in ants" in its current form for publication in Royal Society Open Science. The comments of the reviewer(s) who reviewed your manuscript are included at the foot of this letter.

on behalf of Prof Kevin Padian (Subject Editor)
openscience@royalsociety.org

Associate Editor Comments to Author:

Thank you for submitting this revised paper, which is now ready for acceptance.
Congratulations.

Reviewer comments to Author:

Reviewer: 1

Comments to the Author(s)

The authors put a lot of effort into improving the manuscript, it was a pleasure to read it. This work expands the knowledge about another aspect of social insects social immunity. I have no longer any reservation about its publication.

Reviewer: 2

Comments to the Author(s)

The authors performed a careful revision.

Appendix A

Reviewer Comments

As was stated, one of the experimental series (Control vs. prey with high amount of dry conidia (HC)) was performed one year later than the rest and with entirely different colonies. These might have been sampled in the same region as the former used colonies, which could indicate a high level of relatedness between colonies, but there is still the possibility, that, for example the different time of colony collection (fall 2017 vs. spring 2019) could affect the ants behaviour. Some of the results obtained with these colonies seem to stick out of the observed pattern, at least when looking at some graphs. I understand that sometimes due to circumstances some experiments will be performed later than the rest of the study, which might also involve the testing of new animals. However, this should then in some way be incorporated into the statistical tests performed. Unless the authors can give a strong reasoning about treating and comparing these results statistically the same, I have reservations about rendering the study and the outlook solid. I would therefore suggest to analyse the results using a model (GLM or GLMM) that includes 'Colony' as a factor. This would give clear testament as to whether the four different colonies used influenced the observed behaviours.

The analysis of results from the first experimental series revealed that ants did not avoid a prey contaminated with fungal spores. Thus, we thought relevant to complete these data with a second set of experiments in order to see whether the "no avoidance" response persisted when ants encountered a prey covered with a very huge amount of spores (close to the amount of spores released by a sporulating corpse). Since this second set of experiments was carried out later, we agree with referee 1 that there is a possibility for some colony/time effect. However, we tried to limit as much as possible such an effect by collecting colonies at the same location. Furthermore, before testing the HC condition, we checked that the 4 new colonies (second series) showed a similar level of response to fungus contamination as the first 6 colonies (first series). To this aim, we carried out a few complementary experiments on the 4 new colonies to check whether the ants behaved similarly by retrieving a prey covered with a low amount of conidia (Ctrl-LC ; n=60 ants tested) and by avoiding a sporulating fly (Ctrl-Spo; n=20 ants tested). We found that the retrieval rate of infected prey did not differ between the two series for both conditions (Chi² test: with S1=45.5% and S2=35.5% for Ctrl-LC condition: $\chi^2=0.53, df=1, p=0.46$; S1=3.8% and S2=0% for Ctrl-Spo condition: $\chi^2<0.001, df=1, p=1$). Moreover, the percentage of ants contacting one or both prey was similar between the two series (Chi² test: with S1=43.6% and S2=53.3% for Ctrl-LC condition: $\chi^2=1.10, p=0.29$; with S1=22.7% and S2=15% for Ctrl-Spo condition: $\chi^2=0.19, p=0.66$). Finally, the time spent on the contaminated prey was not significantly different between the two experimental series for each condition (Mann-Whitney: for Ctrl-LC condition: U=3746.5, p=0.14 ; Ctrl-Spo condition: U=830.5, p=0.047). Put all together, the results show that ants from the two experimental series responded in a similar way when being face to the same sanitary threats, i.e. showing a clear avoidance of sporulating flies and by retrieving preys covered with a low amount of conidia. This information is now included in the "Methods" section of the revised MS line :244-257 and recaps in the table S.1.

As suggested by referee 1, we carried out GLMMs analyses with "Colony" as a random factor in order to analyse the followings variables: (1) the overall total percentage of prey retrieved, (2) the percentage of ants contacting one or both prey, (3) for ants contacting only one fly, the percentage of prey retrieved by ants (4) for ants contacting both prey the percentage of no-choice ants. The other variables (i.e. number of contacts, time spent contacting the prey) were not analysed using GLMM due to an important underdispersion of data that we could not handle by changing link function or by data transformation. In these latter case, we used non-parametric statistics. Accordingly, changes were made in the result section at lines 313-333-350-386 as well as in the "data analysis" section

I also have a number of more detailed comments I outline below.

Abstract

Page 2, line 39: You write, that there were six established conditions, and then name only 4. Please include the other 2. Without this information it was rather confusing to follow the results summarized in the latter sentences in the abstract.

As formulated, we agree with referee 1 that the sentence is confusing. The other two conditions were the “two control flies” and the “decaying prey vs sporulating fly”. We re-edited the sentence in order to better explain the tested conditions of prey choice.

As I understand from the abstract, the authors artificially subjected prey items to the entomopathogen fungus, to which the ants did not react with avoidance of the prey items. Latter they write that the ants avoided sporulating flies and freshly dead flies were caught less. Does this mean that the non-avoidance of prey items was just for dry conidia (be it low or high concentration)? I would consider rewriting a bit to emphasize that while all infection was artificial, artificial covering means dry conidia.

Actually, we found out that *M.rubra* did not avoid prey items that were covered by dry conidia (at low but also high concentration). Avoidance and lower catching rate were observed for sporulating prey and for prey freshly killed by the fungus respectively. Following referee comments, we fully re-edited the abstract in order to better emphasize on the main results of our study as well as on the artificial procedure of fungal infection.

Of notice, we talked about “dry conidia “ because we let the solvent of conidia suspension evaporate out of the prey body surface for 1 hour but, on further thoughts, we now prefer to use the simple term “conidia” in the revised MS.

Page 2, line 42: How were the flies naturally infected with *Metarhizium*?

Thanks for the comment. Actually, we now make clear that all the infection were artificial since spores were topically applied with a syringe or through a vortexing procedure.

Page 2, line 44-45: Do the ants really not detect dry conidia presence or are they, in this instance capable, to remove dry conidia by cleaning the item and so can make use of the prey to feed the colony? Leaf-cutting ants, by hitchhikers riding on the leaves back to the nest, clean their leaf fragments of pathogens to reduce pathogen load before they reach the nest.

Referee 1 raises a very interesting point. On video recordings, it was difficult to accurately quantify any licking behaviour of the prey item and close by-eye observation somewhat disturbed the tested ant. One may assume that a licking behaviour expressed by the ant would result in a higher duration of contacts with the prey in our tests. However, the contact duration on prey did not differ in the presence of conidia (LC or HC) or not (Ctrl) as shown for ants contacting one prey (see lines 359 or both prey (see lines 408 and fig6). This suggests that ants did not invest time in spore removal at the site of food discovery. Nevertheless, we cannot exclude that a prey covered with spores would be cleaned before entering the nest or before being fed to nestmates inside the nest. The question you raised is currently under investigation in another study investigating the impact of prey infectiousness on its management by ants inside the nest. This point is now discussed at lines 442-447 in the revised MS.

Introduction

Page 4, line 67: You write that evidence for parasite pressure is lacking, because they have developed effective defences at the group level. I would like to disagree with the authors' reasoning here and argue, that the development of defence mechanisms, for example grooming behaviour or the production of antipathogenic substances in the ants glands is evidence for the pressure that parasites or pathogens put on the members of the group. Please rewrite.

Referee 1 is perfectly right. This sentence is misleading. Line 55 was reedited.

Page 5, line 115: As this paragraph outlines avoidance behaviour of social insects to compromised food sources, I would like to suggest also these references to include, as they relate directly to the topic of the manuscript, both showing research of 'food' avoidance as well as food cleaning behaviour in leaf-cutting ants foraging on life plant matter. Coblenz & Van Bael 2013, Field colonies of leaf-cutting ants select plant materials containing low abundances of endophytic fungi. *Ecosphere*, 4, Article 66 Griffith and Hughes 2010, Hitchhiking and the removal of microbial contaminants by the leafcutting ant *Atta colombica*, *Ecological Entomology* 35, 529-537 Vieira-Neto et al. 2006. Hitch-hiking behaviour in leaf-cutter ants: an experimental evaluation of three hypotheses. *Insectes Sociaux*, 53, 326-332

Thank you for suggesting these additional references. After reading them, we decided to add these citations in our MS to emphasize the ability of some ant species to detect and avoid the retrieval of infected items inside the nest. Line 101-105

Page 5, line 125: ...shared with individuals;

Page 5, line 136: hemocoel

We made the changes.

Material and Methods

Page 5, line 147: Were these colonies queenright or queenless? Was any brood collected? When were the colonies collected? Could you also briefly outline the life history of this species (i.e. colony size, nest site, distribution, etc.)

We added these information at lines 136 to 142.

Page 6, line 169-170: 'leads the insect to death' – please rewrite. 'causes the death of the insect' for example.

Thank you, we made the changes.

Page 6, 180-181: This means, that after 2 days you acquired a sporulating corpse, I assume. Freshly dead corpses were washed as described but then immediately used for testing and not put into Petri dishes? For clarity, please state very clearly at which step you would get a freshly dead corpse and when the result was a sporulating one.

Page 7, line 201: Please give this information at the beginning of the Materials and Methods section (as see my question above).

Page 7, line 231-233: Please see my question above (page 6, 180-181) in this regard and include this information earlier in the M&M section.

Freshly dead corpses (either killed due to fungal infection or killed by exposure to cold) were not washed and were immediately used for testing. Only the corpses that were used at the

sporulating stage of fungus development (or that were used as a “decaying prey” control) were washed before being put for two days in a thermostatic cabinet under conditions favouring sporulation. This “washing step” is made according to the method of Lacey (Manual of techniques in insect pathology. San Diego, Calif.: Acad. Press; 1997.595 409 p. (Biological techniques series)) and aims to limit the development of other pathogens over the corpse during the incubation. This method was commonly used in other studies (e.g.: Pull et al., 2018, in eLIFE). We re-edited this section to improve clarity, namely by providing this information in the first part of the M&M section **at lines 170-172** before detailing the different tested conditions.

Page 7, line 189: homogenised; Page 7, line 203: of food; page 7, line 206: consisted of

Thanks for the suggested changes

Page 7, line 215: Do you have any indication how high or low these control flies are in their pathogen load?

We checked that control flies were not naturally covered with a high amount of conidia. To this aim, before carrying out the first experiments, we took 20 flies that were reared on the same substrate as the tested flies. These flies were washed in a vial containing 50 µL Triton X 0.05 solution per fly and after vortexing the solution, we collected 10 µL that we observed under a microscope with a Thoma's cell. We did not find any fungal conidia on those flies. The pathogen load of control flies is thus negligible (in particular with respect to the conidia-covered prey (low or high)).

Page 8, line 243: Here you write that you tested 136 ants under control prey – sporulating prey (Ctrl – Spo) conditions. In the supplementary material, Table S.1 the total number of ants tested under these conditions (column 6) is 75. Vica versa in line 249 you give 75 as sample size when testing decaying prey vs sporulating prey and in Table S.1 the sample size is given as 136. Please check and change accordingly either in the main body of the text or in the supplementary material.

Thank you very much for highlighting this mistake. We corrected the information in the Material and Method section. We tested 75 ants and 136 ants in the conditions Ctrl-Spo and Dc-Spo, respectively. See **lines: 236 & 241**

Page 8, line 245: When reading I was a bit unclear about the procedure for the decaying prey. Did it go through a sham treatment of conidia (only solvent but no actual conidia), then killed by cold, kept in a petri dish at 25°C for 2 days and then offered as prey? It would be less confusing to just give in a short sentence the information about the whole treatment about how you acquired decayed prey.

Decaying prey were obtained by first killing the insect by exposure to cold. We washed flies cadavers following the method of Lacey, as we did for sporulating flies. Then we kept the insect in a closed Petri dish at 25°C for 2 days before using it in choice experiment.

Page 8, line 250: except

done

Page 8, line 251: I find it a bit worrying that one of the experimental series was performed with other colonies than the other 5 ones. At least you should include a justification why these results can be used in the statistical analysis together with the other 5 experimental series which were all performed with the same colonies.

As explained above, we checked that similar responses were observed between the two experimental series, when being tested under the Ctrl vs LC condition as well as under the Spo vs Ctrl conditions. Since we found no significant differences between these two experimental series, we analysed data from these two experimental series together. This information has been added at lines 244-257 in the revised MS

Page 8, line 255: How was this contact achieved? Antenna, legs or mouthparts?;

We referred “contact” as antennal contacts with the prey when the tested ant headed for the prey. Contact with mouthparts were difficult to see on video recordings and contacts with the legs (e.g. an ant walking near the prey touching it with a leg) were not specifically quantified. We clarified it line 260 & 265.

line 272: I believe there is a word missing at the end of the sentence?

Thank you for your careful reading. We completed the sentence

line 272: performed Dunn’s post-hoc tests; line 273: using the “FSA” package

Thanks for these comments. We made all the suggested changes

Results

Page 9, line 305: See my earlier comments above about treating the series HC, that was performed using entirely different colonies, as if it were the same as the other 5 performed series. Either clearly justify your choice of using these results and why the use of the chosen statistics is still appropriate, or, as I would strongly suggest, use a statistical model (like GLM or GLMM) that incorporates ‘colony’ as a factor. Then you would clearly know whether using different colonies led to the drop in prey retrieval (or other results) even below the percentage of flies killed by fungal infection.

Please see our answer to your first comment

Page 9, line 310: Please give the actual percentages for conditions Ctrl vs. Spo and Dc vs. Spo rather than ‘around 35%

Following your comment, we indicated the actual percentages for these conditions in the MS. Line 320.

Line 314: one of the two prey

The change was done.

Page 9, line 315: How did the ants behave that only chose one prey? Did they walk on a direct path towards the one prey, did they find it after walking around the arena? Please describe a bit the walking/approaching behaviour of the ant towards the fly .

The approaching behaviour was quite the same between the two groups of individuals (those that contacted one prey and those that contacted the two prey). Ants usually walked around the arena and seemed to find a prey only by chance. Thus, we did not notice neither a higher boldness nor a more targeted walk toward the chosen prey among ants that contacted a single prey. Unfortunately, we cannot provide supporting quantitative data since video recordings started only when the ants were close to the offered prey. This by-eye observations would require further experiments to be confirmed.

Page 10, line 329: except; line 336: associated with

Thanks for your careful reading. We made the changes

Page 10, line 354: What about the decaying flies (Dc)? So far you mentioned Ctrl, LC, HC and Spo. How often was decaying prey contacted?

Many thanks for noticing. It was a mere oversight. The number of contacts on the decaying prey was the same as for Ctrl, LC and HC prey.

Page 10, line 355: please put the information regarding grooming behaviour of ants into a separate Paragraph ; line 356: depending on

We made these changes. The new paragraph begins now **line 365**

Page 11, line 374: Please use another word than 'population' for the individuals contacting both prey items.

We replaced it by "groups", **line 380**

Page 11, line 382: In regards to the ants that made a choice; line 405: See above, please use a separate paragraph for grooming behaviour.

We made the suggested changes. The new paragraph began at **line 413**

Page 12, line 424: As you chose to emphasize these two series in regards to retrieval time of noninfected prey when infected prey was nearby, what was the actual elapsed time until retrieval in these two conditions in comparison to other conditions?

All the actual times elapsed until retrieval of control prey (including censored data) were used to draw the survival curves presented in figure 7. To meet referee 1 comment, we added information on the retrieval time of control prey in the Ctrl-HC and Ctrl-Spo condition as well as in the reference condition (Ctrl-Ctrl). We added this information at **lines 430-432**

Discussion

Page 12, line 428: a prey corpse

Done

Page 12, line 441: One could draw this conclusion as a possible explanation as you already formulated with a 'suggests' rather than a 'shows', however, it is also possible that through grooming behaviour, *M. rubra* manages to remove fungal spores and make use of the discovered prey item. While an increase in grooming was not observed in the feeding arena of the experiment, more extensive prey grooming might take place once the prey reaches the nest where it might be effectively cleaned by a group of ants. Is the lack of increased grooming in the experimental arena your only reason why you think the worker could not detect the spore load?

Not only the lack of increased grooming but also the quite unexpected high rate of retrieval of HC prey suggest that ants could not detect the spore load. Having said that, referee 1 is right that ants could manage to remove spores from food items once they are inside the nest and thus could make use of this food source. We added a few sentences on this interesting idea at **lines 442-447**. Please see also our answer to a previous comment on **Page 2, line 44-45**

Page 12, line 443: such as the choice

done

Page 12, line 444: For reference [25], did the authors there not also argue, that excavating a nest in contaminated soil could also be a way to build up a resistance to the encountered, often harmful microorganisms?

Here, our intent was merely to provide other examples of non-avoidance of fungus conidia in other contexts than foraging.

Page 12, lines 444-457: In relation with my comment above, does this section of your discussion not rather explain, that ants might still detect the artificially applied conidia load on dead prey, but because of the described findings and behaviours, the colony can still make use of this type of prey without a high danger of infecting colony members?

We are only suggesting that the sanitary risk associated with the retrieval of a spore-covered prey may be lower than first expected because of the described behaviours. Whether this type of prey can even be used by ant workers as “safe” food for the colony is still unknown to us.

Page 13, line 467: a wide range

Done

Figures

Page 19, Figure 3: What is the reason that you changed the presentation of data in comparison with Figure 2. In both cases you present percentage data of workers performing a task (either prey retrieval or contacting one prey item) or not performing/performing another task (no prey retrieval or contacting both prey items). Both tasks together amount to 100%, composed of grey bar on top of a black bar. In Figure 3, the grey bar is now set to the right instead of on top of the black bar. The data can be displayed either way, I was just wondering whether there was any significance to it? If not, I would think about sticking to one type of figure.

There was no specific reason, so we fully agree that the MS will benefit from sticking to one type of figure. We changed the style of figure 2 to stick to the presentation of figure 3.

Page 19, line 672-673: ‘Control regroups all prey used as control in choice tests.’ I do not understand this sentence.

We simply meant that all control flies (except the decaying prey) were pooled. This is now changed at line 682.

Page 20, line 686: ants retrieving the control prey (or the decaying prey in the last series).

This is now added in the caption at line 698

Page 22, line 698: retrieved Page 22, line 700: Red: Ctrl vs Ctrl. You should also either use abbreviations for every series, as you did for the first three series or the formulated words as you did for the last two series.

We made the requested changes and used abbreviations for every series (line 716)

Bibliography

Please take care to put all the species names in italics. Also, edit for citation style, don't use a mix of either starting with a major letter and the rest in minor, all nouns start with a major letter,

all words are written in major letters. I suppose this is the result of using Mendeley/Zotero/Endnote where this has not been edited.

We have to apologize for the poor editing of the reference section. This is actually the result of using Zotero. We made a careful check of the references.

We would like to thank referee 1 for his/her comprehensive and insightful review of our paper that was really helpful to improve the MS.

Reviewer: 2 Comments to the Author(s) The authors tested *Myrmica* ants for avoidance behavior towards fly prey at different stages of fungal pathogen infection. The authors show that the ants do not discriminate against prey covered in conidia, but chose healthy over infected prey at later stages of infection, particularly at the stage of sporulation, when the ants also perform more selfgrooming after prey handling. The experiments are well-designed, the data clearly presented and discussed.

The paper reads very well, except for the abstract . 1) In the abstract, it is unclear what the authors mean with artificial vs natural infection (in fact, both are artificial, but the first is covering a healthy prey in fungal conidia, vs the other is a later stage of disease).

Referee 2 is right. We no longer speak about "natural infection"

"prey" is missing after "sporulating". Also, the last half sentence is highly speculative and has not been addressed in the manuscript and hence should be removed

2) The references from 34 onwards are all shifted by 2, which needs to be corrected.

Actually, we fully re-edited the whole section of references

3) Technically, topical application of conidia by a Tween or Triton X (were really both used?) should not be referred to as "dry spores" (l 183), which would reflect application of dry conidia from agar plate in the absence of a surfactant.

Thank you very much for your comment. We used the term "dry conidia" because the conidia suspension, which was topically applied on the prey, was allowed to air dry during 1 hour before the beginning of the experiment. Then, no solution remained on the insect cuticle. However, following your comment, we decided to remove the term "dry conidia" and to replace it by "conidia".

Actually, both Tween and Triton X solutions were used but not simultaneously. To obtain a conidia suspension, we first washed sporulating cadavers in Triton X 0.05% vial. After a quick vortexing session, we centrifugated the vial during approximately five minutes in order to create a pellet of spores. We removed the cadavers and the Triton X solution carefully without touching the pellet and then we re-added Tween 20 0.05% solution in the vial. Then we homogenize the suspension by vortexing and disaggregating the pellet. Finally, we assessed the concentration of spores with a Thoma's cell using a microscope.

4) l 272 sentence incomplete

Thanks. We have now completed line 281

Reviewer: 3 Comments to the Author(s) This manuscript represents a comprehensive collection of behavioral experiments in which *Myrmica rubra* ants were allowed to forage for *Drosophila* cadavers that were recently killed, decaying, covered in fungal spores of *Metarhizium brunneum* or killed by infection with this fungal species. The authors performed choice experiments in which ants showed a significant difference between interaction with and collection of sporulating cadavers and all other offered fly types. The data was analyzed using appropriate statistical methods and accompanied

by both a comprehensive introduction and discussion. This makes me conclude that this work is acceptable for publication as is.

My only suggestion to the authors is that they might want to look at their data for the individual ants and how they were showing either more or less risky behavior with regards to the prey that they were presented with. The authors found individual variation, which is very interesting. This makes me wonder if the potential scouts were more risk-taking (aka more exploratory) while the regular foragers might have been less so. Is there a way for the authors to backtrack if perhaps the first ants entering the foraging arena (the potential scouts) were introducing the seen variation in individual behavior versus ants that might have shown up later to forage for prey?

Referee 3 raised a very good point. One may assume that scouts (the first ones to enter the test arena) would be more risk taking than the following ones. Following your comment, we re-analysed data by comparing the three first arriving ants (so-called "scouts" and the three last arriving ants (so-called "foragers"). In total, the sample was 102 ants in each group ("Scouts" or "foragers"). The percentages of ants retrieving an infected prey were similar for the "scouts" and the "foragers" in the Ctrl-LC (33% vs 38%), in the Ctrl-HC (42% vs 33%), Ctrl-FKill (17% vs 17%), Ctrl-Spo (0% vs 0%) and Dc-Spo (0% vs 0%). In all conditions, Chi² tests were not significant. Thus, in our tests, there was no evidence for a "more risky" behaviour of the first tested ants compared to the later arriving ones. In order not to lengthen the MS, we decided not to put this information in the revised version. Of course, we can add these data if requested by the referee/editor.

Furthermore, they might want to look at the following lines and consider re-writing them for clarity or checking them for perceived typos:

Line 91: consider revising "fungi whose some strains"

See line 78

Lines 96-99: very long and difficult to follow sentence.

See line 83-85

Consider revising. Line 306: "fly" probably needs to be "flies"

Line 314: add the word "of" to the end of the line to "one of the two prey"

Thanks for your careful reading of the MS. We made the suggested changes